# Catalytic carbon–carbon bond cleavage in lignin via manganese–zirconium-mediated autoxidation

Chad T. Palumbo[1], Nina X. Gu[1], Alissa C. Bleem[1], Kevin P. Sullivan[1], Rui Katahira[1], Lisa M. Stanley[1], Jacob K. Kenny[1,2], Morgan A. Ingraham [1], Kelsey J. Ramirez [1], Stefan J. Haugen[1], Caroline R. Amendola[1], Shannon S. Stahl [3,4] ✉ & Gregg T. Beckham [1] ✉

Efforts to produce aromatic monomers through catalytic lignin depolymerization have historically focused on aryl–ether bond cleavage. A large fraction of aromatic monomers in lignin, however, are linked by various carbon–carbon (C–C) bonds that are more challenging to cleave and limit the yields of aromatic monomers from lignin depolymerization. Here, we report a catalytic autoxidation method to cleave C–C bonds in lignin-derived dimers and oligomers from pine and poplar. The method uses manganese and zirconium salts as catalysts in acetic acid and produces aromatic carboxylic acids as primary products. The mixtures of the oxygenated monomers are efficiently converted to *cis,cis*-muconic acid in an engineered strain of *Pseudomonas putida* KT2440 that conducts aromatic *O*-demethylation reactions at the 4-position. This work demonstrates that autoxidation of lignin with Mn and Zr offers a catalytic strategy to increase the yield of valuable aromatic monomers from lignin.

Lignin is the most abundant natural source of aromatic units, comprising 15–30 wt% of lignocellulosic biomass[1,2], and many contemporary efforts are pursuing lignin valorization via depolymerization to aromatic monomers[1,3–7]. Multiple depolymerization strategies have been developed that cleave aryl–ether C–O bonds very effectively[3–6,8]; however, the various C–C linkages in lignin, formed during its biosynthesis[9] or from condensation reactions during lignocellulose processing[6,10], are much more difficult to cleave[11].

To date, very few catalytic methods have been reported to cleave C–C inter-unit linkages in lignin. A notable example uses a Ru/NbOPO₄ catalyst that cleaves both C–O and C–C bonds and yields deoxygenated lignin oils with up to 32 wt% monomers. This system generates monocyclic products in significantly higher yields (up to 153 mol%) than those accessible from nitrobenzene oxidation, a method that

primarily cleaves C–O linkages[12,13]. Wang et al. reported that up to 13 wt% of phenol can be obtained through tandem C–O and C(aryl)–C bond cleavage using a multifunctional catalyst comprising CuCl₂/AlCl₃ and Ru/CeO₂[14]. Similarly, Han and colleagues reported an 8 wt% yield of phenol with a zeolite catalyst[15]. Stahl and colleagues reported oxidative catalytic fractionation of lignin affording C–C cleavage products[16], and Waldvogel and colleagues showed vanillin could be obtained by oxidation of kraft lignin with peroxodicarbonate[17]. Other methods may generate products of C–C cleavage, but mechanistic information is lacking, the monomers are formed yields below values that can be obtained through C–O cleavage alone, or the processes require multiple steps and proceed in lower yields[18–21].

Lignin oil from reductive catalytic fractionation (RCF) is an ideal substrate to explore C–C bond cleavage in lignin due to the near-

[1]Renewable Resources and Enabling Sciences Center, National Renewable Energy Laboratory, Golden, CO 80401, USA. [2]Department of Chemical and Biological Engineering, University of Colorado Boulder, Boulder 80303 CO, USA. [3]Department of Chemistry, University of Wisconsin-Madison, Madison, WI 53706, USA. [4]Great Lakes Bioenergy Research Center, University of Wisconsin-Madison, Madison, WI 53706, USA. ✉e-mail: stahl@chem.wisc.edu; Gregg.Beckham@nrel.gov

theoretical C−O bond cleavage that occurs during this process, resulting in an oil wherein the dimers and oligomers are only linked by C−C bonds (Fig. 1)[10,22–24]. Recently, Samec and coworkers demonstrated that a TEMPO+-derived oxidant [TEMPO= (2,2,6,6-tetra-methylpiperidin-yl)−1-oxyl] is able to produce 1.9 mmol of 2,6-dimethoxybenzoquinone/g of oligomers derived from RCF of birch wood[25]. The reaction uses 400 wt% of TEMPO+ oxidant with respect to the lignin-derived substrate, although the oxidant could be regenerated electrochemically and re-used to afford similar product yields in subsequent cycles.

Despite these promising reports, there remains a need for catalytic systems that can cleave C−C bonds in lignin to utilize lignin more effectively and offer product flexibility for lignin valorization to useful products. One option is autoxidation, a radical-chain process that is widely used in the commodity chemical industry (e.g., in the production of phenol from cumene[26], cyclohexanol/cyclohexanone[27], terephthalic acid and other aromatic carboxylic acids[28–32], and alkyd paints[33,34]). Co, Mn, and other metal ions are often used as catalysts in these processes, leveraging their ability to break down hydroperoxides into reactive alkoxyl radicals via the Haber-Weiss reaction[28,31,35]. The resulting alkoxyl radicals can undergo C−C bond cleavage via β-scission. Seminal reports from Partenheimer demonstrated production of aromatic carboxylic acids from 3,4-dimethoxytoluene, as a simple lignin model, and from various lignin sources using Co/Mn/Br and Co/Mn/Zr/Br catalyst systems, respectively[36,37]. In two recent studies, we demonstrated C−C bond cleavage in mixed plastics and an acetyl-protected poplar RCF oligomer substrate using a Co/Mn autoxidation catalyst system paired with a N-hydroxylphthalimide or bromide co-catalyst, respectively[38–40].

In an attempt to develop catalyst systems that use more abundant and sustainable metals, here we present an autoxidation catalyst for C−C bond cleavage that avoids the need for cobalt[41]. Further, we sought to eliminate bromide as it is corrosive and necessitates the use of expensive titanium-clad reactors due to its corrosivity[29]. Studies comparing oxidations of an acetyl-protected lignin model demonstrated Mn is not active enough for appreciable C−C bond cleavage. However, we observed C−C bond cleavage with Mn when studying the methyl-protected analog, 4-propylveratrole, in the presence of a Zr co-catalyst. Thus, in this work we use a homogeneous Mn and Zr autoxidation catalyst system to cleave the C−C bonds in oligomers of methylated pine (primarily G-type lignin) and poplar (G- and S-type lignin) RCF oils, along with the model compounds 1−6 shown in Fig. 2. We demonstrate that this Mn/Zr catalyst system produces a mixture of bio-available monomers that can be converted to cis,cis-muconic acid using biological funneling[38,42–45], a strategy that utilizes genetically engineered microbes to convert mixtures of compounds into a single target product, with a strain engineered to demethylate the methyl-stabilized phenol[42,46]. An overview of the workup procedure from biomass to cis,cis-muconic acid is shown in Supplementary Fig. 1.

## Results

### Methyl protection of phenolic groups enables lignin autoxidation with Mn and Zr only

We first explored oxidations of mono-aromatic model compounds with propyl side chains to test whether C−C bond cleavage can be achieved using Mn-catalyzed autoxidation, relative to the Co/Mn/Br system previously studied, which cleaves the C−C bonds of acetyl-protected lignin aromatics[40]. As we will present below in more detail, we observed C−C bond cleavage products with methyl-protected substrates in the presence of a Zr co-catalyst. The "standard conditions" shown in Fig. 3 are those determined to be optimal for the RCF oligomer substrate (vide infra), and the yields for model compound experiments are reported in mol% with respect to the number of aromatic rings needed to produce the product. The oxidation mixtures were derivatized by methylation prior to analysis using gas chromatography with a flame ionization detector (GC-FID) for accurate quantification of the carboxylic acids (Supplementary Figs. 2 and 3; Supplementary Tables 1 and 2). The model compounds and analytical standards were synthesized as detailed in the ESI (Supplementary Figs. 4–27). The complete distribution of products of the reactions in Figs. 3 and 4, as discussed in further detail below, can be found in Supplementary Figs. 28 and 29 of the ESI.

As a negative control reaction, we initially attempted to oxidize 4-propylguaiacol, **1-OH**, a coniferyl (G)-type monomer, as shown in Fig. 3. Treatment of **1-OH** at 6 bar $O_2$ and 150 °C with 8 mol% $Mn(OAc)_2 \cdot 4H_2O$ and 6 mol% $Zr(acac)_4$ (acac = acetylacetonate) as catalysts in a steel reaction vessel resulted in 94(1)% conversion (hereafter denoted using 94(1)% notation) of **1-OH** to several products, but only 2.1(6) mol% of vanillin was obtained as a result of C−C bond cleavage. The product of C−C bond coupling, divanillin, was produced in 11(3) mol% yield.

Due to the low recovery of aromatic products with **1-OH**, attributed to the reactivity of phenolic compounds under oxidizing conditions, we next sought oxidations of lignin models with phenol-stabilizing groups. The attempted oxidation of the acetyl-protected analog of **1-OH**, namely 4-propylguaiacyl acetate, **1-Ac**, with a Mn catalyst resulted in 4.6(1)% conversion of the substrate and no appreciable yield of C−C bond cleavage products. When Zr was added, 18(1)% of the substrate was converted, but products were produced at only 1.9(7)%. This demonstrates that both Mn and Mn/Zr are less active catalysts than our previously reported Co/Mn/Br system[40], which converts 82(10)% of the 4-propylguaiacyl acetate in 2 h at 120 °C and produces 47(8)% of products through C−C bond cleavage[40].

We hypothesized that changing the phenol-stabilization group to one that is more electron-donating would enable the Mn catalyst to oxidize the lignin model substrate. Consequently, we targeted the methyl-protected analog of 4-propylguaiacol, **1-OH**, 4-propylveratrole, **1**, along with 1-(3′,4′,5′-trimethoxyphenyl)propane, **2**, the methyl-protected analog of the syringyl (S)-type 4-propylsyringol, as shown in Fig. 3. Treatment of **1** under the standard oxidation conditions yielded 30(8) mol% of products comprising 13(7) mol% of veratric acid, 5.5(3) mol% of veratraldehyde, 5.1(9) mol% of 2-acetoxypropioveratrone, and 6.2(5) mol% of 2-methoxymaleic acid, the lattermost evidently arising from ring opening of **1**. Similarly, **2** exhibited a total product yield of 48(2) mol%, including a 14.5(6) mol% yield of 3,4,5-trimethoxybenzaldehyde and 23(1) mol% yield of 3,4,5-trimethoxybenzoic acid. Like the reaction of **1**, the 2-acetoxy aryl ketone was also produced, in 10.5(5) mol% yield. In the absence of the Zr co-catalyst known to improve autoxidation selectivity and prevent $MnO_2$ precipitation oxidation[28,47,48], **1** yielded only trace quantities of products.

### The aromatic acid and aldehyde products exhibit limited stability under catalytic conditions

Subjecting the S- and G-type aromatic aldehyde and carboxylic acid products to the standard conditions resulted in reduced aromatic product recovery, as shown in Fig. 4. For example, veratric acid was

**Fig. 1 | Representative lignin C−C bonds.** β-1, β-5, β-β, and 5-5 carbon–carbon bond linkages found in oligomers of RCF oil.

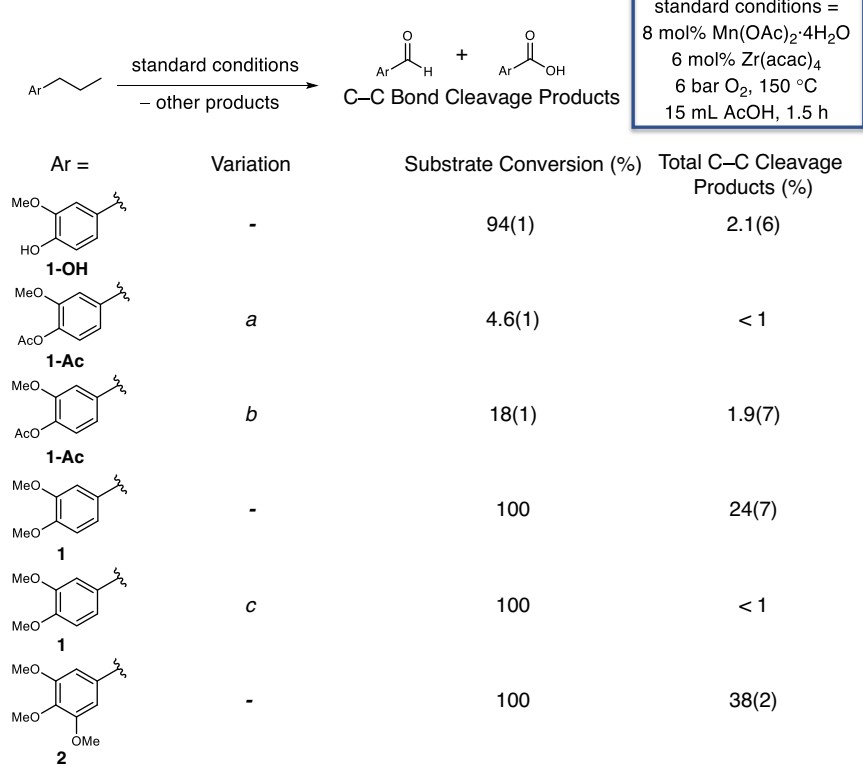

**Fig. 2 | Lignin models.** Structures of model monomers and dimers **1**–**6** were used in this study.

**Fig. 3 | Monomer model oxidations.** Autoxidation of monomer models **1-OH**, **1-Ac**, **1**, and **2**, with yields shown as mean (standard deviation) in mol% aromatics. Standard conditions: substrate, 0.1 mmol; catalyst, 8 mol% Mn(OAc)$_2$•4H$_2$O, 6 mol% Zr(acac)$_4$; solvent, 15 mL acetic acid; O$_2$ loading, 6 bar; time, 1.5 h; temperature, 150 °C. [a]no Zr(acac)$_4$ and reacted for 2 h. [b]reacted for 2 h. [c]no Zr(acac)$_4$.

recovered in 62(7) mol%, whereas veratraldehyde was recovered in 38 mol% (5(1) mol% veratraldehyde and 33(4) mol% as veratric acid). Similarly, 3,4,5-trimethoxybenzoic acid was recovered in 51(12) mol%, and 3,4,5-trimethoxybenzaldehyde was recovered in 43 mol% (20 mol% aldehyde and 23 mol% as 3,4,5-trimethoxybenzoic acid). Each substrate produced 2-methoxymaleic acid, which is indicative of oxidative ring-opening.

**Dimer model compound oxidation reactions with the Mn/Zr catalyst system demonstrate monomer production through C−C bond cleavage**

In a similar manner, the methylated dimers **3**–**6** (Fig. 2) were oxidized using the standard reaction conditions, as shown in Fig. 5. Notably, we

discovered that this catalyst system liberates monomers from 3 of the 4 types of C−C bonds studied (β−1, β−5, β-β).

The model β−1 dimer, 1,2-bis(3,4-dimethoxyphenyl)ethane, **3**, when oxidized, gave a total product yield of 35(4) mol% aromatics consisting of 29(3) mol% of veratric acid, 2(2) mol% veratraldehyde, and 4.4(4) mol% of 2-methoxymaleic acid. Oxidation of the β-β model compound eudesmin, **4**, afforded a total product yield of 45(2) mol% comprising 37(2) mol% veratric acid, 4.4(1) mol% veratraldehyde, and 2.9(2) mol% methoxymaleic acid. The β−5 dimer, **5**, gave a total product yield of 30(1) mol% of veratric acid, 12.5(8) mol%, veratraldehyde 15(1) mol%, 2-methoxymaleic acid 2.0(4) mol%, along with trace quantities of 3,4-dimethoxyisophthalic acid.

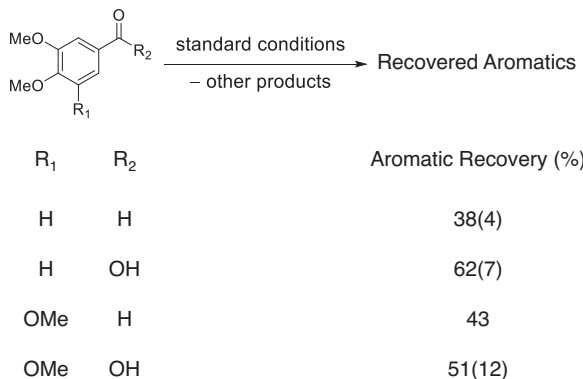

| $R_1$ | $R_2$ | Aromatic Recovery (%) |
|---|---|---|
| H | H | 38(4) |
| H | OH | 62(7) |
| OMe | H | 43 |
| OMe | OH | 51(12) |

**Fig. 4 | Product stability studies.** Product stability reactions under autoxidation conditions with yields shown as mean (standard deviation) in mol% aromatics. Standard conditions: substrate, 0.1 mmol; catalyst, 8 mol% Mn(OAc)$_2$•4H$_2$O, 6 mol% Zr(acac)$_4$; solvent, 15 mL acetic acid; O$_2$ loading, 6 bar; time, 1.5 h; temperature, 150 °C.

The 5-5 dimer model, 1,1'-dipropyl-3,3',4,4'-tetramethoxybiphenyl, **6**, did not yield monomer products in significant quantities and afforded 30(1) mol% of products comprising 13.7(7) mol% of the dimethyl divanillate, 7.6(4) mol% of dimethyl divanillin, and 6.9(4) mol% of the mixed acid/aldehyde analog. Additionally, 2.1(1) mol% of the monocyclic 3,4-dimethoxyisophthalic acid was identified, evidently due to ring cleavage of one of the aromatic units.

### RCF oil methylation followed by distillation separates monomers and produces an oligomer substrate for autoxidation

RCF was used to produce lignin oils from pine and poplar using a 5 wt% Ru/C catalyst in methanol at 225 °C for 6 h with 30 bar H$_2$ (Supplementary Figs. 30–34; Supplementary Tables 3–7)[49]. The total yields of monomers were quantified by GC-FID to be 1.5 mmol/g of phenolic pine RCF oil, and 1.8 mmol/g of phenolic poplar oil. Phenol stabilization by methylation of the RCF oils was accomplished by stirring the oils in acetonitrile with excess potassium carbonate and methyl iodide at room temperature for 60 h. For pine, the monomers were recovered quantitatively as their methylated analogs. For poplar, near-quantitative recovery was observed for all monomers except the S-type monomers 4-propenyl-2,6-dimethoxyphenol (44% recovery) and 3-(4-hydroxy-3,5-dimethoxyphenyl)−1-propanol (8% recovery). Residual phenols in the methylated oils were not detected by $^{31}$P NMR spectroscopy (Supplementary Fig. 35 and Supplementary Table 8).

Distillations of the methylated oils were performed at 220 °C and 10 mbar for 30 min. The GC-FID quantification data of monomers in the oligomer and distillate fractions from the pine distillation showed a monomer recovery of 98% in the distillate collection flask with <1% remaining in the bottoms. The distillation of methylated poplar oil also gave high recoveries: 95% of the monomers were recovered with only 3% detected in the bottoms fraction. The gel permeation chromatography (GPC) data are consistent with the quantification data and the successful removal of monomers to produce oligomer-enriched substrates for both pine and poplar, as shown in Fig. 6. A combination of GPC, GC-FID, and gas chromatography-mass spectrometry (GC-MS) data on the pine distillate and analytical standards shown in Fig. 6b and Supplementary Fig. 33 indicated that dimers were present in the distillate.

### Autoxidative C−C cleavage affords monomers at 28 mol% of the RCF yields

We next sought to optimize the autoxidation catalysis conditions using the methylated pine oligomer substrate, including reaction temperature, oxygen loading, catalyst loading, substrate loading, reaction time, and Zr loading, as shown in Fig. 7. As done for the model compounds, the oxidation mixtures containing carboxylic products were derivatized by methylation prior to GC-FID analysis (ESI Methods). The yields are reported in mmol analyte/g of substrate in the bar charts with wt% (total g pre-derivatized analytes/g substrate x 100) yields reported above each bar. Numerical values for the optimization conditions are presented in Supplementary Tables 9–14.

A screen of six reaction temperatures (Fig. 7a) ranging from 90–190 °C showed a distribution of yields that increased sharply from 0.23 mmol/g of oligomer substrate at 130 °C to the maximum yield of 1.2 mmol/g at 150 °C. At 170 °C, the yield decreased to 1.1 mmol/g and to 0.84 mmol/g at 190 °C. The oxygen partial pressure also affected monomer yields (Fig. 7b). At $P_{O2}$ = 0 bar, no monomers were detected, as expected. There was an increase in total monomer yield from $P_{O2}$ = 1 bar, which gave a yield of 0.86 mmol/g of oligomer substrate, to 1.2 mmol/g at 6 bar. An oxygen loading of $P_{O2}$ = 8 bar did not improve the overall yield. An increase in total monomer yield was also observed with increased catalyst loadings (Fig. 7c) and substrate concentrations (Fig. 7d). When changing the catalyst loading while maintaining a substrate quantity of 50 mg, the total product yields increased from 0.032 mmol/g of pine oligomer substrate with 0 wt% catalyst to 0.16 mmol/g with 4 wt% Mn(OAc)$_2$•4H$_2$O and 6 wt% Zr(acac)$_4$. Using 8 wt% Mn(OAc)$_2$•4H$_2$O and 12 wt% Zr(acac)$_4$, a substantial increase in the product was observed (1.00 mmol/g of oligomer substrate). Increasing catalyst loadings further did not produce significantly higher product yields. Similarly, while maintaining a constant catalyst loading of 8 wt% Mn(OAc)$_2$•4H$_2$O and 12 wt% Zr(acac)$_4$, an increase in the total monomer yield was observed from 0.63 mmol/g with a 1.3 mg/mL concentration of substrate to 0.92 mmol/g with a 4.3 mg/mL concentration of substrate. A time course study demonstrated an increase in the total monomer yield from 0.71 mmol/g at 0.17 h to 0.92 mmol/g at 0.5 h (Fig. 7e). For reaction times between 0.5 h and 1.5 h, similar yields ranging 0.91–1.1 mmol/g were observed. Longer reaction times did not increase the total monomer yield. Lastly, we studied the effect of Zr loading. As shown with **1** in Fig. 3, the presence of Zr(acac)$_4$ improved the yield of products (Fig. 7f). Similarly, when Zr(acac)$_4$ was omitted from the oligomer oxidation reaction, only 0.061 mmol/g of identifiable products were quantified (primarily veratric acid and aldehyde). With a loading of 6 wt% Zr(acac)$_4$, the overall yield increased to 0.77 mmol/g, and 12 wt% yielded 0.90 mmol/g. No substantial increase was observed with 24 wt% Zr(acac)$_4$.

Using the optimized conditions identified from the experiments presented in Fig. 7, a total monomer yield of 1.01(4) mmol/g of oligomer substrate was obtained for the oxidation of the methylated pine oligomer substrate, as shown in Fig. 8a–c. For poplar, we applied the same conditions and obtained 0.96(2) mmol/g, as shown in Fig. 8d–f. The 1.01(4) and 0.96(2) mmol/g total monomer yields obtained from pine and poplar, respectively, each represent 28 mol% of the total monomer yield of their respective RCF process. A representative calculation for pine is presented in Eqs. (1) and (2) below. Numerical values of the autoxidation yields are presented in Supplementary Tables 15−16.

$$\frac{1.01\,\mathrm{mmol\ monomer}}{g\ \mathrm{pine\ oligomer\ substrate}} \times \frac{0.40\,g\ \mathrm{pine\ oligomer\ substrate}}{g\ \mathrm{pine\ methylated\ RCF}} \times \frac{1.065\,g\ \mathrm{pine\ methylated\ RCF}}{g\ \mathrm{pine\ phenolic\ RCF}}$$
$$= \frac{0.43\,\mathrm{mmol\ monomer}}{g\ \mathrm{pine\ phenolic\ RCF}}$$

$$\tag{1}$$

$$\left(\frac{0.43\,\mathrm{mmol\ monomer}}{g\ \mathrm{pine\ phenolic\ RCF}}\right) \Big/ \left(\frac{1.52\,\mathrm{mmol\ monomer}}{g\ \mathrm{pine\ phenolic\ RCF}}\right) \times 100 = 28\% \tag{2}$$

### Engineered bacteria enable the conversion of pine oxidation mixtures to *cis,cis*-muconate

Aromatic-catabolic microbes can catabolize heterogenous mixtures of aromatic compounds toward a single target product through a process

**Model Dimer Oxidations**

**Fig. 5 | Dimer model oxidations.** Autoxidation of dimer models **3**−**6** with yields shown as mean (standard deviation) in mol% aromatics. Standard conditions: substrate, 0.1 mmol; catalyst, 8 mol% Mn(OAc)$_2$·4H$_2$O, 6 mol% Zr(acac)$_4$; solvent, 15 mL acetic acid; O$_2$ loading, 6 bar; time, 1.5 h; temperature, 150 °C.

termed biological funneling[38,42–45]. The oxidative catalytic approach described above generates a slate of low molecular weight, water-soluble products, which are ideal for use as microbial growth substrates[50]. Here, strains of *Pseudomonas putida* KT2440 (hereafter, *P. putida*) were engineered to produce *cis,cis*-muconic acid from the aromatic monomers in the oxidation streams of methylated pine oligomers[51]. As shown in Fig. 8a, veratric acid and veratraldehyde comprised a large fraction (0.76 mmol/g substrate) of these streams, but these *p*-methoxylated compounds are not natively assimilated by *P. putida* (Supplementary Fig. 36). Fortunately, a cytochrome P450 monooxygenase from *Rhodopseudomonas palustris* HaA2 catalyzes *p*-demethylation of veratrate[52], so this operon (CYP199A4-HaPux-HaPuR) was genomically integrated into *P. putida* under control of the strong P$_{tac}$ promoter with optimized ribosome binding sites for each gene, generating strain ACB236 (Supplementary Table 18). Additionally, preliminary experiments indicated that the native vanillate demethylase, VanAB, could *O*-demethylate veratrate to isovanillate (Fig. 9A and Supplementary Fig. 37), so the isovanillate demethylase operon *ivaAB*[53]

was integrated to prevent undesirable accumulation of isovanillate, resulting in *P. putida* strain ACB262. The ESI Methods and Supplementary Tables 17–20 provide further details on strain construction and genotypes.

To ensure that *P. putida* could utilize the major constituents of the oxidized RCF oligomer stream, strain ACB262 was cultivated in an M9 minimal medium with 2 mM veratrate or 2 mM veratraldehyde as the sole source of carbon and energy. In both cases, the model compound was fully consumed within 50 h, and no accumulation of isovanillate was observed (Supplementary Fig. 38a, b). Next, strain ACB262 was evaluated for its ability to grow with oxidized oligomer streams as the sole source of carbon and energy. To prepare substrates for growth experiments, methylated oligomers of pine RCF oil were oxidized in triplicate at the 65 mg scale, treated with an aqueous base to precipitate catalyst, neutralized, and filter sterilized (ESI Methods)[38]. The resulting solutions were added at 10% v/v to M9 minimal medium, with initial media compositions provided in Supplementary Table 21. When *P. putida* ACB262 was grown in M9 with 10% v/v of oxidized RCF

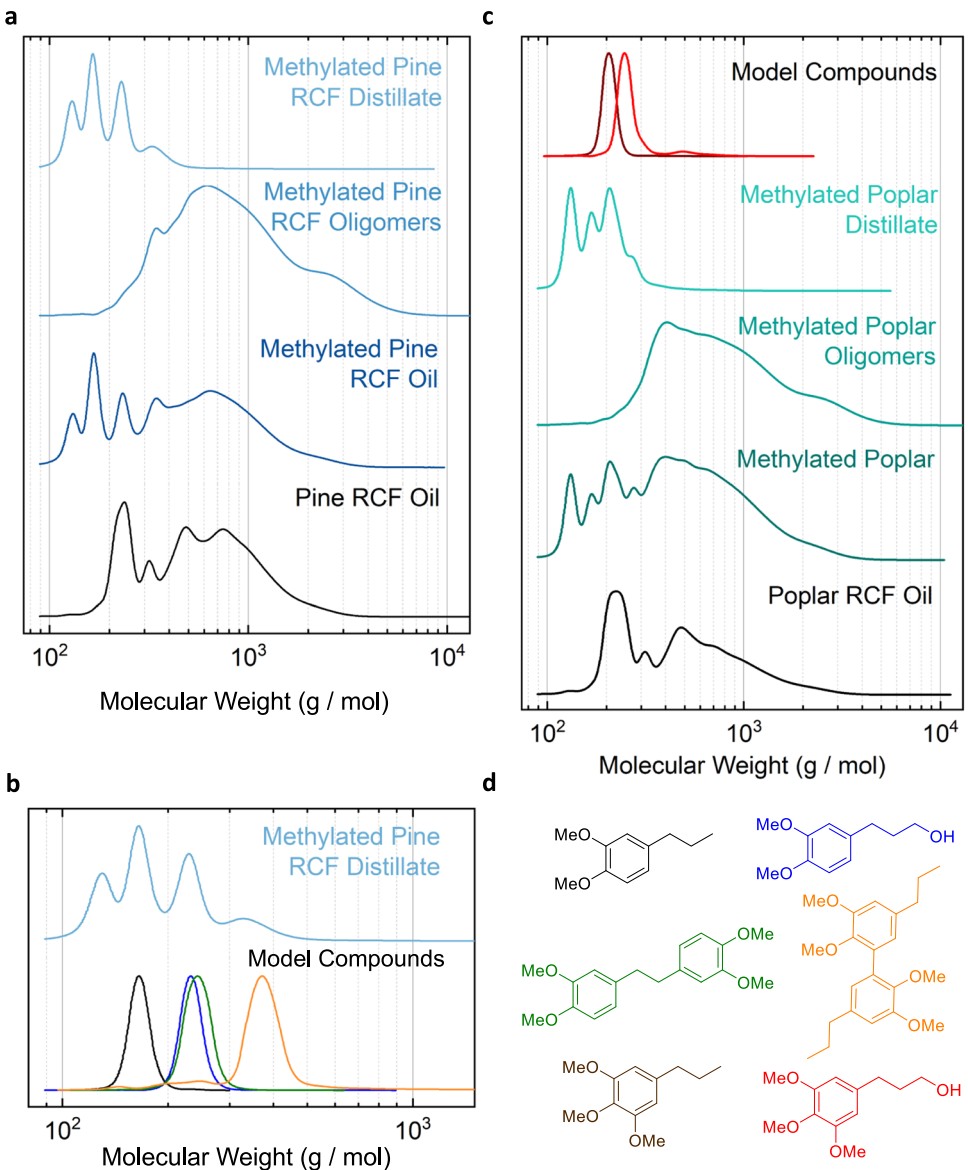

**Fig. 6 | Substrate GPC characterization. a** GPC traces of the pine RCF substrate (black), methylated pine RCF (dark blue), methylated pine oligomers (blue), and methylated pine distillate (light blue). **b** Comparison of the GPC traces of the pine distillate (light blue) with those of 4-propylveratrole (black), **1**, propanol veratrole (blue), 1,2-bis(3,4-dimethoxyphenyl)ethane (green), **3**, and 2,2′,3,3′-tetramethoxy-5,5′-dipropyl−1,1′-biphenyl (orange), **6**, indicating dimers present in the distillate.

**c** GPC traces of the poplar RCF substrate (black), methylated poplar RCF (dark green), methylated poplar oligomers (green), methylated poplar distillate (light green), 1-(3′,4′,5′-trimethoxyphenyl)propane (brown), **2**, and 1-(3′,4′,5′-trimethoxyphenyl)propan-3-ol (red). **d** Molecular structures of models compounds in panels **b**, **c**.

oligomers, it simultaneously utilized both veratrate and veratraldehyde from these streams without the need for a supplemental carbon source and without accumulation of intermediates (Supplementary Fig. 38c–e).

Additional metabolic engineering was required to produce muconate from veratrate and veratraldehyde in *P. putida*. A previously described strain, CJ781[51], was modified to include the optimized CYP199A4-HaPux-HaPuR and *ivaAB* operons for *p*-demethylation as described above, as well as the mutant AroY$_{E474V}$ for decarboxylation of protocatechuate to catechol (Fig. 9A)[54]. The genotype of the resulting *P. putida* strain, ACB263, is shown in Supplementary Table 20. To verify functionality of the engineered pathway, strain ACB263 was grown with the model compounds veratrate and veratraldehyde, and glucose was added to 5 mM every 24 h to support growth. Veratrate and veratraldehyde were both consumed within 48 h

and fully converted to muconate at 90.4(7)% and 96(2)% yields, respectively, with minimal accumulation of intermediates (Fig. 9B, C). Strain ACB263 was also cultivated with the same oxidized RCF oligomer streams as ACB262, and it fully converted the major monomer constituents (veratrate, veratraldehyde) to muconate within 24 h, achieving 100(2)% yield from these two substrates (Fig. 9D and Supplementary Fig. 39). Importantly, uninoculated control media incubated under the same conditions did not exhibit considerable changes in composition (Supplementary Fig. 40), indicating that all measured changes in analytes were associated with the action of the bacteria. Of note, *p*-methoxybenzoate was detected in small amounts in the oxidation streams of methylated pine RCF (Fig. 8a), and it is a viable substrate for the CYP199A4 system[52]. However, *p*-methoxybenzoate was not analyzed because its contribution to the total muconate yield was assumed to be negligible in these dilute biological preparations.

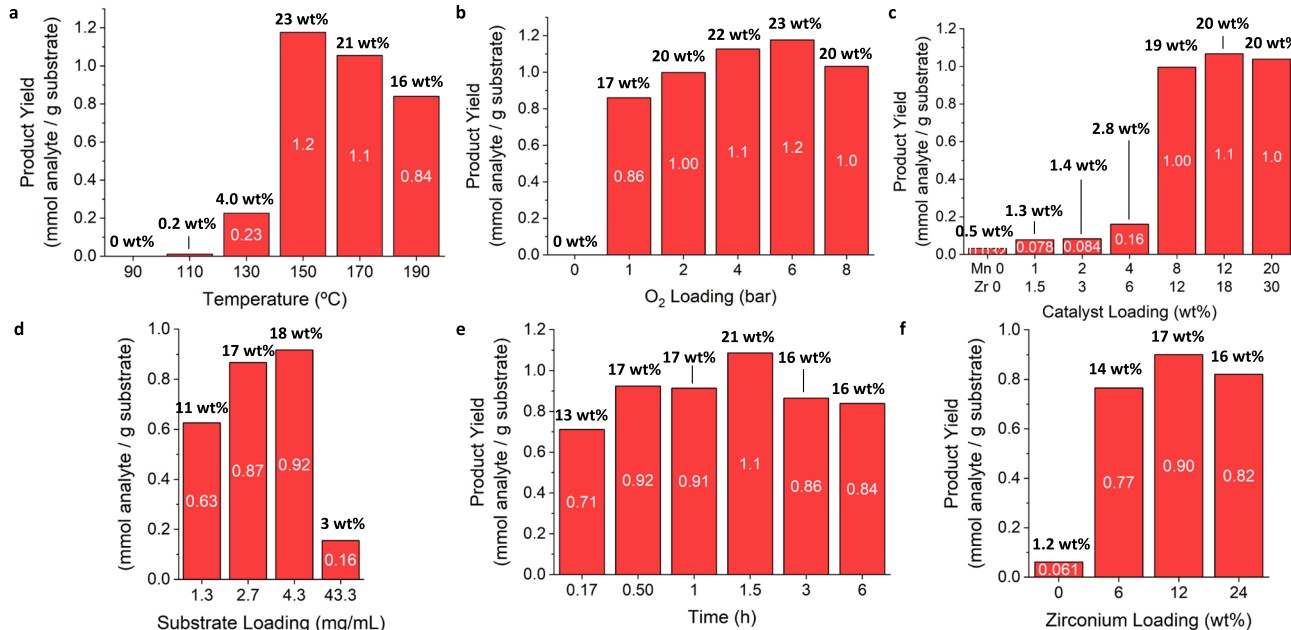

**Fig. 7 | Optimization of conditions on pine RCF oligomers.** GC-FID quantification of optimizations of **a** temperature, **b** O₂ loading, **c** catalyst loading, **d** substrate loading, **e** time, and **f** zirconium loading for the autoxidation catalysis of methylated pine RCF oligomer oxidation. Conditions except the individual parameter changed: substrate, 50 mg; catalyst, 8 wt% Mn(OAc)₂·4H₂O, 12 wt% Zr(acac)₄; solvent, 15 mL acetic acid; O₂ loading, 6 bar; temperature, 150 °C. Numerical values for the individual products are presented in Supplementary Tables 9−14.

## Discussion

These results demonstrate the catalytic production of bio-available monomers through C−C bond cleavage, as shown unambiguously using oligomers prepared from pine and poplar RCF oils, and their biological conversion to a single product, *cis,cis*-muconic acid. This Mn/Zr autoxidation catalyst system allows cleavage of the aryl propane units in 3 of the 4 types of carbon−carbon linkages present in lignin, shown in Fig. 1 (β-1, β−5, β-β).

While oxidations of lignin models and lignin have been studied with Co/Mn/Br and Co/Br catalysts[36,37,55], the target of those studies was not C−C bond cleavage. Rather, the goal was to increase monomer yields from the oxidation of β-O-4-containing substrates by acetylation of the generated phenolic products in situ prior to their degradation under the oxidation conditions. Conversely, our study presents a method to overcome the inherent limit of monomer yields from C−O cleavage alone by catalytically cleaving C−C bonds in lignin oligomers. This is accomplished by effecting β-scission of the high-energy radical intermediates generated via autoxidation. We further describe methods that enable the identification and quantification of a mixture of oxygenated products along with a gene editing protocol that enables its biological conversion using engineered strains of *Pseudomonas putida*.

While promising, the overall process performed in batch reactors is limited by the need for stabilization chemistry and limited product stability, as observed in our model compound oxidations in Figs. 3−5. Although autoxidation catalysis for lignin warrants phenol stabilization, there are several options that could facilitate progressing this chemistry further. One approach to improve the viability of lignin methylation could be to use a flow system with dimethyl carbonate as a methylating reagent[56−59]. Further, the regeneration of dimethyl carbonate from carbon dioxide is an active area of study and would enable a circular methylation process[60−62]. Flow chemistry could also be used to overcome issues with aromatic decomposition. The products could be continuously removed from the reaction in a flow reactor, which could enable lignin oxidation without the need to separate out the initial monomers generated from RCF. Alternatively, if the RCF monomers with propyl chains are sufficiently valuable, other separation methods could be employed including advanced chromatographic techniques and membrane separations[63−66] that allow monomer separation and preparation of RCF dimers and oligomers.

Mechanistic details of Co/Mn/Br/Zr-mediated autoxidations in acetic acid have been studied for decades, but to date, there is not a universally accepted model. This is likely because kinetic parameters vary substantially with the reaction conditions and substrate[67−70]. Dimeric and trimeric acetate complexes of Co(III) and Mn(III) have been proposed as intermediates[71−74], and differences in the activity between various Co(III) species have been observed spectroscopically along with the decay of active Co(III) to more stable forms[28,74−77]. Less reactive forms have been proposed to exist as bridged hydroxo Co(III) dimers[75−77], as well as hydroxo-substituted oxo-centered trimers[74]. But their interconversion with less hydroxylated structures at elevated temperatures has been described by varying water concentrations[71]. Regarding Zr, its role in the oxidation of RCF oligomers is not obvious based on product yields and distributions. In the industrial Co/Mn/Br mediated autoxidation of alkylbenzenes, Zr is used as a co-catalyst, especially for the oxidation of poly(alkyl)benzenes[28]. Several hypotheses based on kinetics data involving Co have been proposed to explain the increase in activity when Zr is added[28,30,47,68,78−81]. Chester et al. proposed from kinetics models that Zr forms heterobimetallic complexes with Co(III), thereby preventing decay to more stable Co(III) complexes[78]. Partenheimer speculated that Zr affects the selectivity of toluene oxidation by promoting hydroperoxide coordination and dehydration to benzaldehyde and preventing the precipitation of MnO₂[47,48]. While it is unclear whether these models can be extrapolated to our Mn/Zr catalyst and lignin substrate, we did observe an increase in selectivity toward reactions that produce the desired acid and aldehyde products as opposed to deleterious side reactions. It should be noted that Partenheimer also observed a selectivity difference in oxidations of hydroxymethylfurfural. With Zr, double the yield of 2,5-diformylfuran was observed[82].

The Br co-catalyst in Co/Mn/Br also differs significantly from our Mn/Zr catalyst. In the former system, Br initiates the radical-chain process by H-atom transfer. In the absence of Br, it has been proposed that Mn(III) generates radicals by single-electron transfer from the

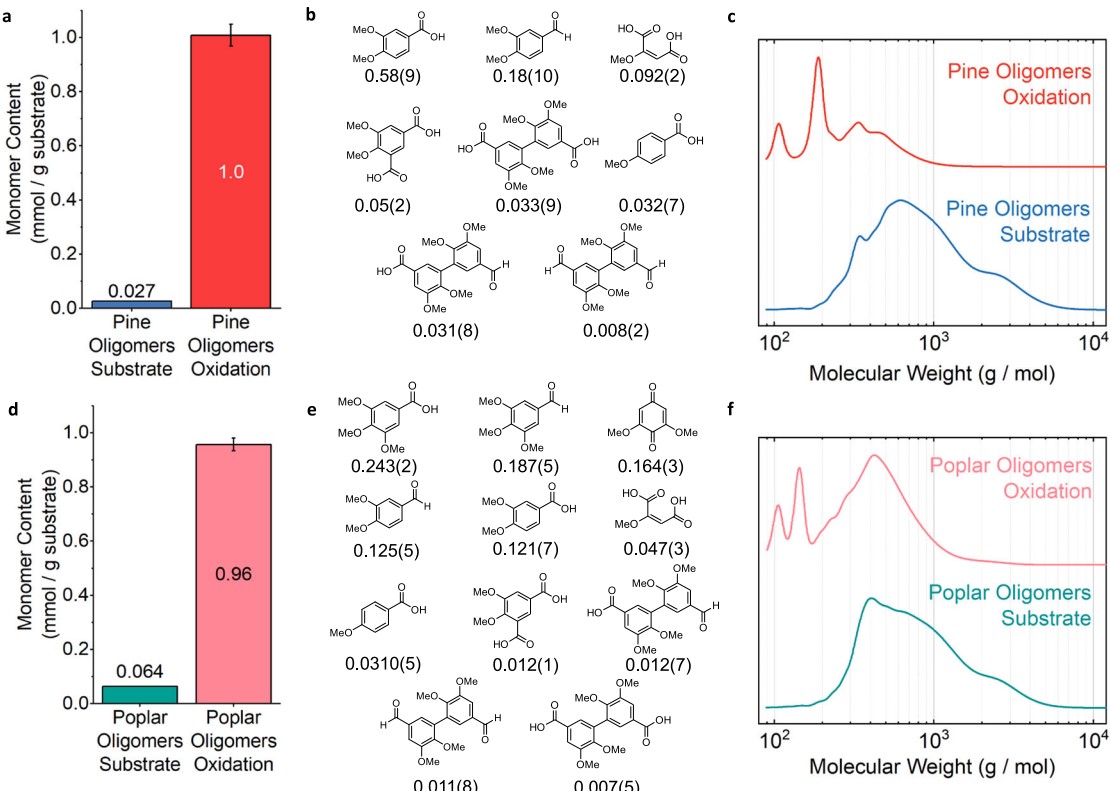

**Fig. 8 | Optimized oligomer oxidation. a** GC-FID quantification of total monomers in the pine oligomers substrate (blue) and oxidation mixture (red) in mmol/g substrate. Error bars indicate the standard deviation from the mean of three replicates. **b** Oxidation products identified and quantified in the pine oligomer oxidation (yields in mmol/g substrate). **c** GPC data comparing the molecular weight distribution of the pine oligomers substrate (blue) and pine oligomers oxidation (red). **d** GC-FID quantification of total monomers in the poplar oligomers substrate (green) and oxidation mixture (pink). Error bars indicate the standard deviation from the mean of two replicates. **e** Oxidation products identified and quantified in the poplar oligomer oxidation (yields in mmol/g substrate). **f** GPC data comparing the molecular weight distribution of the poplar oligomers substrate (green) and poplar oligomers oxidation (pink). Numerical data for this fig. are provided in Supplementary Tables 15 and 16. Conditions: substrate, 65 mg; catalyst, 8 wt% $Mn(OAc)_2 \cdot 4H_2O$, 12 wt% $Zr(acac)_4$; solvent, 15 mL acetic acid; $O_2$ loading, 6 bar; time, 1.5 h; temperature, 150 °C.

electron-rich lignin aromatics via the loss of a proton from a radical cation intermediate[83]. It has been suggested that Co(III) is generated from Co(II) in $Co/O_2/AcOH$ systems by adventitious peroxide[84].

Oxidation specifically allows the production of bio-available compounds that can be funnelled into a single product, in this case, *cis,cis*-muconic acid, a platform chemical used in the production of biopolymers including nylon, nylon derivatives, and polyethylene terephthalate (PET), as well as performance-advantaged composites[85–92]. Translation of the CYP199A4 system from *R. palustris* HaA2 provides a catabolic pathway for *p*-methoxylated aromatics in engineered *P. putida*, and the use of oxidized RCF oligomers offers an example of biological funnelling of a heterogenous, lignin-derived stream toward a single product. The oxidized RCF oligomer streams were provided to *P. putida* at dilute concentrations, but the strains rapidly utilized the major monomer constituents (veratrate and veratraldehyde), indicating that tolerance to and conversion of higher substrate loadings is within the performance capability of these strains, as we have demonstrated for similar feedstocks and engineered pathways[93,94]. Although our system does not produce appreciable monomers from 5-5 dimers, a metabolic pathway for cleavage of the 5-5 C–C bond has been reported[95–97]. The resulting monomers—equivalents of vanillate and 4-carboxy-2-hydroxypenta-2,4-dienoate (CHPD), in the case of 5,5′-dehydrodivanillate cleavage—could then be directed toward central metabolism or engineered pathways for muconate production in *P. putida*[97,98]. Going forward, bioprocess development could be combined with metabolic engineering or evolution strategies to improve the performance of these biocatalysts,

increasing the yield of muconate or similar products[93,94,99–102] from lignin.

We have demonstrated C–C bond cleavage in lignin using a Mn and Zr catalyst system in acetic acid, as shown with oligomers derived from pine and poplar lignin RCF oils. The method uses molecular oxygen as the oxidant and phenol-stabilization chemistry and yields of 1.01(4) and 0.96(2) mmol of monomers/g of oligomer substrate were obtained for pine and poplar, respectively. The oxygenated and bio-available product stream produced from pine oligomer oxidation was converted in biological systems using *P. putida* KT2440 to *cis,cis*-muconate.

## Methods
### Autoxidation catalysis of the oligomer fractions
For a single reaction, a 75 mL Parr batch reactor fit with a glass liner insert was charged with 20-65 mg of functionalized RCF oil oligomers, acetic acid (15 mL), a stir bar, and $Mn(OAc)_2 \cdot 4H_2O$ and $Zr(acac)_4$ catalyst. The mixture was pressurized three times with pure nitrogen and subsequently charged with air and nitrogen to achieve the desired oxygen partial pressure. The vessel was heated to temperature, at which, it was maintained for the desired timeframe before it was cooled rapidly in an ice bath. The resultant homogeneous solutions were stored in the freezer until needed for analysis. For analysis, an aliquot of the autoxidation reaction mixture was taken, and the acetic acid was evaporated under a stream of $N_2$ at room temperature. Dimethylformamide, excess potassium carbonate (10 mass equiv.), and excess methyl iodide (10 mass equiv.) were then added to the

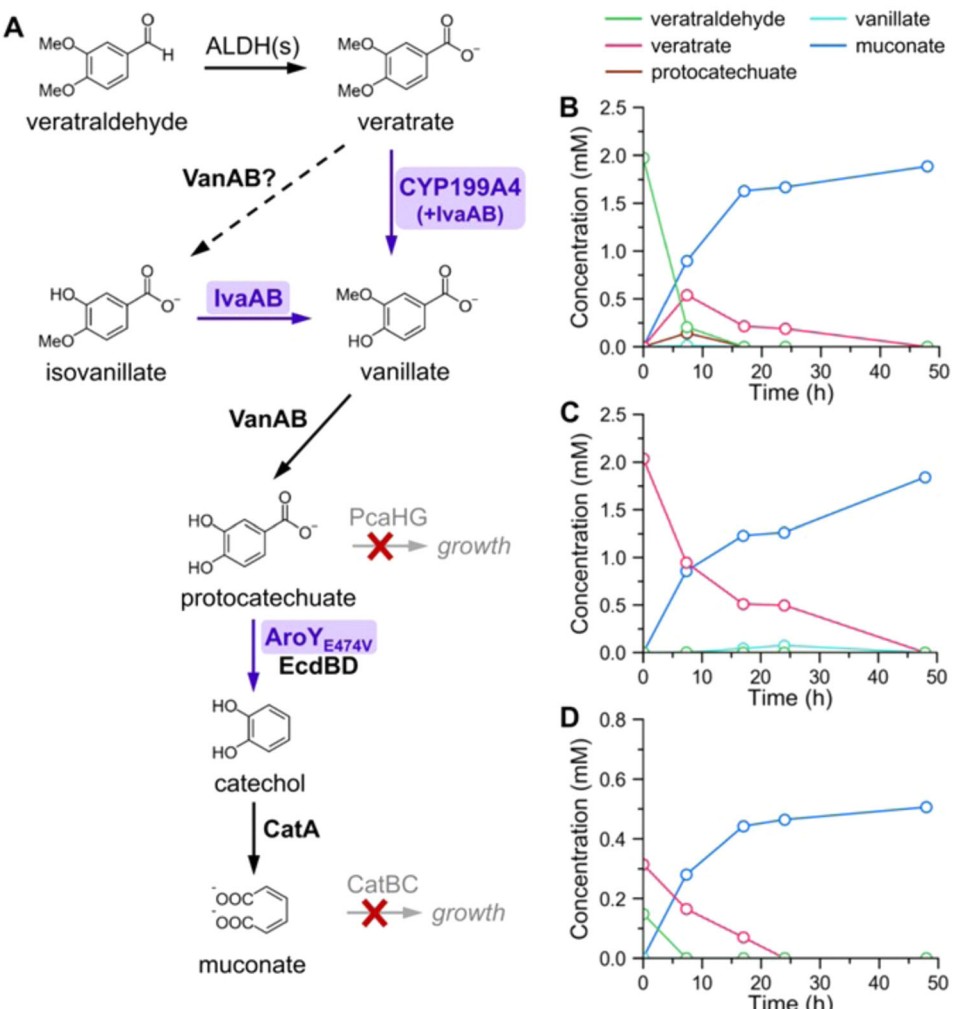

**Fig. 9 | Metabolic pathway and cultivation. A** Metabolic pathway for the production of muconate from oxidation streams of methylated pine RCF in *P. putida* strain ACB263. Strain ACB263 is derived from another muconate-producing strain, CJ781[51], and modifications are indicated by purple boxes/arrows. Aldehyde dehydrogenase(s) (ALDH) catalyze the oxidation of veratraldehyde to veratrate, which then undergoes *p*-demethylation to vanillate by the CYP199A4 system. VanAB *O*-demethylates vanillate to protocatechuate, which is decarboxylated to catechol by the AroY$_{E474V}$-EcdBD system, followed by ring cleavage to produce muconate. Undesirable accumulation of isovanillate was prevented by the isovanillate demethylase, IvaAB. Diversion of metabolites to growth was avoided by deletion of *catRBC* and *pcaHG*. **B** Cultivation of strain ACB263 in M9 + 2 mM veratraldehyde produces the equivalent molar yield of muconate. **C** Cultivation of strain ACB263 in M9 + 2 mM veratrate produces the equivalent molar yield of muconate. **D** Strain ACB263 produces muconate from veratrate and veratraldehyde when grown in M9 + 10% v/v oxidation products from methylated pine RCF (representative plot from one oxidation reaction; two additional preparations are shown in Supplementary Fig. 39). Catechol and isovanillate were not detected in any of the culture supernatants. Error bars indicate the standard deviation from the mean of three biological replicates (error bars are smaller than the size of the marker in most cases; see numerical data for this fig. in Supplementary Table 22).

residue, and the mixture was stirred at room temperature for 3 h. To the slurry, an internal standard was added (naphthalene), and the mixture was filtered with a 0.2 μm porosity filter frit into an analysis vial before being analyzed using the GC-FID methods described in the Supplementary Information.

### Reporting summary
Further information on research design is available in the Nature Portfolio Reporting Summary linked to this article.

## Data availability
Additional synthetic and spectroscopic details, GC-FID quantification data, and strain and biological conversion details are included in this paper and its Supplementary Information files. Any further information is available from the corresponding author upon request.

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

## Acknowledgements

This work was authored in part by the National Renewable Energy Laboratory, operated by Alliance for Sustainable Energy, LLC, for the U.S. Department of Energy (DOE) under Contract No. DE-AC36-08GO28308. Funding to C.T.P., K.P.S., S.J.H., K.J.R., L.D.S., D.G.B., J.B., and G.T.B. was provided by the U.S. Department of Energy Office of Energy Efficiency and Renewable Energy Bioenergy Technologies Office. Funding to N.X.G. and A.C.B. was provided by The Center for Bioenergy Innovation, a U.S. DOE Bioenergy Research Center supported by the Office of Biological and Environmental Research in the DOE Office of Science. Contributions by S.S.S. were supported by the US Department of Energy, Office of Basic Energy Sciences, under award no. DEFG02-05ER15690. The views expressed in the article do not necessarily represent the views of the DOE or the U.S. Government. The U.S. Government retains and the publisher, by accepting the article for publication, acknowledges that the U.S. Government retains a non-exclusive, paid-up, irrevocable, worldwide license to publish or reproduce the published form of this work or allow others to do so, for U.S. Government purposes. We thank David G. Brandner, Jeremy R. Bussard, and Justin B. Sluiter for providing extractives-free pine and poplar RCF oil and Mikhail O. Konev for helpful discussions.

## Author contributions

Conceptualization, C.T.P., N.X.G., A.C.B., K.P.S., S.S.S., G.T.B.; data curation, C.T.P., A.C.B.; formal analysis, C.T.P., A.C.B., M.A.I., K.J.R., S.J.H.; funding acquisition, G.T.B., S.S.S.; investigation and methodology, C.T.P., N.X.G., A.C.B., K.P.S., J.K.K., R.K., L.M.S., C.R.A., S.S.S., G.T.B.; supervision, G.T.B.; visualization, C.T.P.; writing original draft, C.T.P., G.T.B.; review and editing, all authors.

## Competing interests

C.T.P., N.X.G., K.P.S, S.S.S., and G.T.B. have filed a patent application on the catalysis presented in the work. A.C.B., C.R.A., and G.T.B. have also filed a patent application on the *P. putida* strains presented in this work. The remaining authors declare no competing interests.
