## [Peer Review File · Nature Communications]

Catalytic carbon–carbon bond cleavage in lignin via manganese–zirconium-mediated autoxidationREVIEWER COMMENTS

Reviewer #1 (Remarks to the Author):

This paper of Beckham and coworkers addresses an important and timely question in lignin depolymerization: the selective cleavage of C-C bonds in lignin, which is important to increase the overall obtainable value from lignin towards well defined products.

While there are a few systems in recent literature addressing C-C bond cleavage, many of those studies rely on higher temperature transformations, and a somewhat brute-force methods. Here the already known metal-catalysed autooxidation process, that is used in commodity chemicals industry. Applying such approach to lignin C-C bond cleavage is elegant, yet not trivial. Given the presence of phenols in lignin, and derived depolymerization mixtures, one would not necessarily think of using this approach given the possible incompatibility with phenolic moieties. The authors solve this by phenol-protection chemistry both with acetate and methyl capping groups, which allows the chemistry to function beautifully and high selectivity. Some other notable features are the comprehensive approach by which separations are integrated with the catalytic processing. And an important result is then, that the authors target a final single product of higher industrial relevance by biological funneling. Overall, this contribution is excellent and this reviewer recommends it for publication after some minor adjustments, listed below:

Introduction/state of the art: The careful checking of recent literature is recommended, to not miss any recent C-C bond cleavage manuscripts.

Page 3: the authors 'hypothesize, that changing the phenol stabilization group to one that is more electron donating' would make a difference. I wonder whether the marked difference of the methyl protected models is due to electronic effects. Phenolic esters are much more labile both for hydrolysis and towards oxidation compared to their stable ether analogues. The reactivity difference may find its origin rather in the stability of the ethers produced.

To the previous point: simple mechanistic studies may shine light on the stability of the acetate-protected models under these conditions, and would give a more exact explanation of the differences observed. A short discussion to this issue would be useful.

Fig 3, scope: model compound 1 versus model compound 2: the yield of the cleavage products is substantially different. can the authors give more explanation?

Overall mechanism/role of Z : The reviewer acknowledges that the catalytic system is known for decades, yet mechanistic details are still not clarified. It may be equally difficult to come up with a deeper mechanistic view - is this something worth investigating in the future?

Supplementary figure 21, 22: the compound seems to have impurities in the NMR spectra, please note and possibly quantify purity.

Supplementary figure 29-30: spectra appear not properly phase/baseline corrected

Reviewer #2 (Remarks to the Author):

The authors describe a method to convert methylated lignin oil (oligomers from reductive catalytic fractionation of lignin) to muconic acid, using a combination of catalytic C–C bond cleavage (oligomers to carboxylic acid monomers) and bacterial conversion routes (carboxylic acid monomers to muconic acid). This topic is certainly interesting, but the key novelty of this manuscript needs to be strengthened. For example, bacterial conversion of aromatics into various kinds of value-added products and aerobic cleavage of C–C bonds using homogeneous or heterogeneous catalysts have been reported. If Mn/Zr catalytic system for C–C bond cleavage is really efficient compared to routes reported in the literature, much more work is required to demonstrate it.

1. In the introduction, oxidative and reductive C–C bond cleavage are not comparable. The introduction on C–C bond cleavage should be modified.
2. Autoxidation catalyzed by Co/Mn/Zr/Br in acetic acid has been reported for the conversion of lignin model compounds and even with commercially lignin from Sigma-Aldrich to aromatic monomers (DOI: 10.1002/adsc.200800614). The authors should explain the different reaction mechanisms and explain the high efficiency of their Mn/Zr catalyst.
3. The units for catalyst loading and reaction yield are inconsistent.
4. The manuscript discusses the stability of products in standard reactions, can the author explain the relation between the stability of products and the main point of their work? Is there a way to increase the stability and total reaction yield of the aromatic monomers?
5. The authors suggest more electron donating groups on phenol may increase the reaction yield, this should be demonstrated. Could the introduction of other kinds of protecting groups increase the reaction yield?

6. This work only used methylated lignin oil as a substrate, why not use native or commercially available lignin polymer, which contains less phenolic hydroxyl groups and would be better to utilize?
7. For the oil methylation process, some S-type monomers are difficult to be methylated, while no residual phenols in the methylated oils were detected according to the ³¹P NMR result, why?
8. Bacterial conversion of lignin derived aromatics to muconic acid has already been published by the same authors (<https://doi.org/10.1016/j.ymben.2021.12.010>), and the main results of bacterial conversion is not reported paper. It is not clear what improvement is reported in this manuscript.
9. Since the whole process is complicated, the mass balance of the total process needs to be provided.

At this stage I am not convinced that the work offers sufficient novelty for publication in NC.

Reviewer #3 (Remarks to the Author):

The authors employ the Mn/Zr catalytic system to effectively cleave C-C bonds in lignin-derived dimers and oligomers obtained from pine and poplar via the autoxidation method. This process leads to the production of aromatic carboxylic acid, which can be further transformed into cis, cis-muconic acid using an engineered strain of *Pseudomonas putida* KT2440. The manuscript ingeniously combines chemical catalysis and biocatalysis in the conversion of renewable energy into fine chemicals, showcasing impressive scientific achievement. I recommend publishing this work in Nature Communications with minor revisions as outlined below:

1. The metabolic pathway of the engineered strain *P. putida* KT2440, responsible for the conversion of aromatic compounds into cis,cis-muconic acid, lacks sufficient clarity. I urge the authors to provide a detailed explanation in this regard.
2. On page 6, paragraph 3, line 1, the statement "Optimization of reaction conditions led to a 28 mol% increase in RCF monomer yields" requires a more comprehensive explanation of the methodology employed. The current description is overly simplistic.
3. The authors should discuss the distinguishing catalytic mechanisms between the Mn/Zr catalytic system and the Co/Mn/Br system. Although the author speculates on several models in paragraph 3 of page 11, they fail to validate their applicability to the Mn/Zr catalytic system. If relevant, I suggest including a detailed discussion of these catalytic mechanisms.
4. On page 3, paragraph 4, line 4, please revise the phrase "(S)-type 4-propylsyringol" to "(S)-type 4-propylsyringol."
5. In reference 47, please correct "cis, cis" to "cis, cis."

6. The concept of biological funneling is relatively novel. I recommend briefly summarizing it in the appropriate section of the introduction.

7. Within the discussion section, I suggest emphasizing innovative summaries while reducing the focus on addressing existing problems. This adjustment will better highlight the novelty of this research.

Reviewer #4 (Remarks to the Author):

In this study, the authors extended their tandem approach, previously demonstrated in their Science paper, to leverage lignin valorization. The limited utilization efficiency of lignin oligomers remains a significant obstacle within the lignin community, leading research efforts to yield results that may not be immediately ideal but represent essential advancements towards practical lignin utilization. Consequently, this study addresses a critical challenge in the depolymerization of lignin oligomers to lignin monomers through catalytic cleavage of C-C bonds. Additionally, the authors candidly acknowledge the current limitations of their proposed approach. As a researcher and reviewer within the related fields, I appreciate this honesty, which, in my understanding, does not diminish the novelty of the study but provides readers with valuable insights to construct a comprehensive understanding of this investigation.

Main concerns:

1. The authors reference an unpublished work of their own, which is generally acceptable; however, it is essential to highlight the distinctions between the two studies. At first glance, based solely on the titles, the two studies may appear quite similar. If the primary differentiation lies only in the oxidation catalysts employed for lignin depolymerization, the novelty of the findings may be deemed insufficient for publication in Nature Communications.

2. In the tandem approach towards practical conversion of lignin oligomers, multiple steps are unfortunately omitted or oversimplified.

2.1 The use of MeI as the stabilization reagent may give a poor impression that the synthetic route was designed for an organic chemistry lab. While advancements in synthetic chemistry for lignin depolymerization are appreciated, the lignin community, presently interested in practical lignin utilization, tends to favor valorization pathways illustrated with sufficient proof of viability in real lignin valorization. In this case, dimethyl carbonate is proposed as an alternative, but the references (48 and 49) provided are insufficient to validate the suitability of the reaction for lignin methylation.

2.2 The oxidative depolymerization of lignin oligomers was conducted in acetic acid at a relatively low substrate loading (4.3 mg/mL). According to the findings in Fig. 7d, the product yield increased with higher substrate loading. To gain more practical insights, it is highly recommended to explore the depolymerization reaction at higher substrate loadings, as converting 4.3 g of oligomer samples in 1 L acetic acid would barely be considered in real cases.

2.3 The process of biological funneling using the depolymerized oligomers is not clearly explained. One crucial step is omitted, in which additional details are required to elucidate how the depolymerized oligomers were isolated from the mixture after the oxidation reaction. According to the supplementary experimental information (PS26), the samples (~30 mg in 45 mL acetate acid) were dried under nitrogen at -20 °C. The rationale behind this operation should be carefully elucidated, as it serves as a pivotal step bridging the chemical oxidation and biological funneling processes.

Reviewer #1:

This paper of Beckham and coworkers addresses an important and timely question in lignin depolymerization: the selective cleavage of C-C bonds in lignin, which is important to increase the overall obtainable value from lignin towards well defined products.

While there are a few systems in recent literature addressing C-C bond cleavage, many of those studies rely on higher temperature transformations, and a somewhat brute-force methods. Here the already known metal-catalysed autooxidation process, that is used in commodity chemicals industry. Applying such approach to lignin C-C bond cleavage is elegant, yet not trivial. Given the presence of phenols in lignin, and derived depolymerization mixtures, one would not necessarily think of using this approach given the possible incompatibility with phenolic moieties. The authors solve this by phenol-protection chemistry both with acetate and methyl capping groups, which allows the chemistry to function beautifully and high selectivity. Some other notable features are the comprehensive approach by which separations are integrated with the catalytic processing. And an important result is then, that the authors target a final single product of higher industrial relevance by biological funneling. Overall, this contribution is excellent and this reviewer recommends it for publication after some minor adjustments, listed below:

We thank the reviewer for the positive and constructive feedback! We have attempted to address each of the comments in turn below.

Introduction/state of the art: The careful checking of recent literature is recommended, to not miss any recent C-C bond cleavage manuscripts.

We would be very happy to include specific references suggested by the reviewer. At present, we have included papers from Joseph Samec, Feng Wang, Buxing Han, and others, but certainly we may have inadvertently missed one or more. If the reviewer has specific suggestions, we look forward to receiving that input and we will update the Introduction accordingly.

Page 3: the authors 'hypothesize, that changing the phenol stabilization group to one that is more electron donating' would make a difference. I wonder whether the marked difference of the methyl protected models is due to electronic effects. Phenolic esters are much more labile both for hydrolysis and towards oxidation compared to their stable ether analogues. The reactivity difference may find its origin rather in the stability of the ethers produced.

We appreciate this feedback, and this is an interesting idea. Our sense is that the stability of the ether products obtained in the current work cannot be isolated from the electronic effects of the methoxy substituents. Therefore, one could speculate that the difference in reactivity is attributed to the electronic effects of the methoxy group from methyl protection, which are characterized as more resonance donating than those of acetoxy ring substitution.

At least in the current work using the Mn/Zr system, the idea that the phenolic esters are more labile for oxidation is likely not the case. We tested this directly, as shown in **Figure 3**, where 95% of the phenol ester substrate is recovered, as shown in the second entry, whereas the phenol ether is fully converted to oxidation products.

Figure 3. Autoxidation of monomer models **1-OH**, **1-Ac**, **1**, and **2**, with yields shown as value(standard deviation) in mol% aromatics. Standard conditions: substrate, 0.1 mmol; catalyst, 8 mol% Mn(OAc)₂·4H₂O, 6 mol% Zr(acac)₄; solvent, 15 mL acetic acid; O₂ loading, 6 bar; time, 1.5 h; temperature, 150 °C. *a* no Zr(acac)₄ and reacted for 2 h. *b* reacted for 2 h. *c* no Zr(acac)₄.

To the previous point: simple mechanistic studies may shine light on the stability of the acetate-protected models under these conditions, and would give a more exact explanation of the differences observed. A short discussion to this issue would be useful.

We conducted our model compound oxidation studies of both acetyl- and methyl-protected propyl-guaiacol substrates for this reason. As mentioned above and shown in **Figure 3**, very little reactivity is observed for the acetyl-protected models as opposed to the methyl-protected, which convert fully.

Importantly, we hope we are interpreting the reviewer's comments correctly, and we are happy to consider any other mechanistic studies that the reviewer would suggest.

Fig 3, scope: model compound **1** versus model compound **2**: the yield of the cleavage products is substantially different. can the authors give more explanation?

We thank the reviewer for their attention to detail – this was a mistake that we have now corrected. The total C–C cleavage extent of substrate **1** is within error of substrate **2** and should read “24(7)%”, as originally reported in **Figure S27**. We have fixed this mistake in the updated draft.

Overall mechanism/role of Zr: The reviewer acknowledges that the catalytic system is known for decades, yet mechanistic details are still not clarified. It may be equally difficult to come up with a deeper mechanistic view - is this something worth investigating in the future?

As the reviewer notes, mechanistic details including the involvement of Zr in Co-catalyzed autoxidation reactions have been the subject of study for decades, but models presented in previous works are not cohesive, and it is not clear to us if and how they translate to our Mn/Zr catalyst system presented here. We are actively investigating this with our collaborators at SLAC, and we hope to publish our findings in the

future. In light of this reviewer comment, we have included an expanded discussion on the role of Zr in the Discussion section of the main text. We have also included it below:

Mechanistic details of Co/Mn/Br/Zr-mediated autoxidations in acetic acid have been studied for decades, but to date, there is not a universally accepted model. This is likely because kinetic parameters vary substantially with the reaction conditions and substrate.^{61, 62, 63, 64} Dimeric and trimeric acetate complexes of Co(III) and Mn(III) have been proposed as intermediates,^{65, 66, 67, 68} and differences in the activity between various Co(III) species have been observed spectroscopically along with the decay of active Co(III) to more stable forms.^{25, 68, 69, 70, 71} Less reactive forms have been proposed to exist as bridged hydroxo Co(III) dimers,^{69, 70, 71} as well as hydroxo-substituted oxo-centered trimers.⁶⁸ But their interconversion with less hydroxylated structures at elevated temperatures has been described by varying water concentrations.⁶⁵ Regarding Zr, its role in the oxidation of the RCF oligomers is not obvious based on product yields and distributions. In the industrial Co/Mn/Br mediated autoxidation of alkylbenzenes, Zr is used as a cocatalyst, especially for the oxidation of poly(alkyl)benzenes.²⁵ Several hypotheses based on kinetics data involving Co have been proposed to explain the increase in activity when Zr is added.^{25, 27, 42, 62, 72, 73, 74, 75} Chester *et al.* proposed from kinetics models that Zr forms heterobimetallic complexes with Co(III), thereby preventing decay to more stable Co(III) complexes.⁷² Partenheimer speculated that Zr affects the selectivity of toluene oxidation by promoting hydroperoxide coordination and dehydration to benzaldehyde and prevents the precipitation of MnO₂.^{42, 43} While it is unclear whether all of these models can be extrapolated to the combination of our Mn/Zr catalyst and lignin substrate, we did observe an increase in selectivity toward reactions that produce the desired acid and aldehyde products as opposed to deleterious side reactions. It should be noted that Partenheimer also observed a selectivity difference in oxidations of hydroxymethylfurfural. With Zr, double the yield of 2,5-diformylfuran was observed.⁷⁶

We hope the additional details provided on the role of Zr improves the manuscript and provides further context for the reader.

Supplementary figure 21, 22: the compound seems to have impurities in the NMR spectra, please note and possibly quantify purity.

We thank the reviewer for pointing this out, as we overlooked the small impurities originally in this NMR spectrum. We have quantified the purity of isoelemicin to be 88% based on ¹H NMR spectroscopy with a *p*-xylene internal standard. We have noted this in the SI in the isoelemicin experimental section, and we updated **Figure S33** and **Table S7** accordingly.

Supplementary figure 29-30: spectra appear not properly phase/baseline corrected

We thank the reviewer for bring this to our attention. **Figures S29-30** (now **Supplementary Figs 30-31**) are now properly base/baseline corrected.

Reviewer #2:

The authors describe a method to convert methylated lignin oil (oligomers from reductive catalytic fractionation of lignin) to muconic acid, using a combination of catalytic C–C bond cleavage (oligomers to carboxylic acid monomers) and bacterial conversion routes (carboxylic acid monomers to muconic acid). This topic is certainly interesting, but the key novelty of this manuscript needs to be strengthened. For example, bacterial conversion of aromatics into various kinds of value-added products and aerobic cleavage of C–C bonds using homogeneous or heterogeneous catalysts have been reported. If Mn/Zr catalytic system for C–C bond cleavage is really efficient compared to routes reported in the literature, much more work is required to demonstrate it.

We thank the reviewer for the comments. Here we address first the point about bacterial conversion of lignin-derived aromatic compounds, and second the point about catalytic C–C bond cleavage in lignin.

The main point of the current study is that an industrial oxidation process can be used to cleave C–C bonds in lignin, which is still a considerable challenge. We agree that bacterial conversion of aromatic compounds has been demonstrated previously, including studies from our group, but virtually all previous reports use lignin-based substrates derived from C–O bond-cleavage reactions. The inclusion of a microbial conversion step here is intended to show that monomers obtained from C–C bond cleavage from lignin are bio-available. In addition, for methyl-protected aromatic compounds, additional metabolic engineering can be conducted to enable their convergent catabolism to a single product.

The reviewer's statement that "...aerobic cleavage of C–C bonds ... [has] been reported", is not correct to our knowledge. If the reviewer is aware of references we missed, we would be happy to include them. However, there are very few studies that clearly document C–C bond cleavage in real lignin streams. We believe that we have cited the few relevant examples in the Introduction of our manuscript. The state-of-the-art study was published by Samec and co-workers in *Nature Chemistry* in 2021 using a super-stoichiometric oxidant. This important precedent is the basis for our quantitative comparisons in this report.

Here, we have used an industrial oxidation process with O₂ as the oxidant to show definitively that C–C bonds can be catalytically cleaved, enabling the monomer yield from lignin to exceed the value set by the C–O bond content. For these reasons, we feel that this study is a breakthrough that merits publication in a top tier journal such as *Nature Communications*.

1. In the introduction, oxidative and reductive C–C bond cleavage are not comparable. The introduction on C–C bond cleavage should be modified.

We are unsure of what the reviewer means. In the Introduction, and as pointed out by Reviewer 1, we attempted to be inclusive of the various routes that have been employed for C–C bond cleavage in lignin. We fully recognize that reductive and oxidative strategies for this purpose are quite different, both mechanistically and often practically (e.g., reductive heterogeneous catalysis and oxidative homogeneous catalysis), but we feel that these approaches should be inclusively cited. If the reviewer has concrete suggestions for how to modify the Introduction or additional citations to include, we would be happy to consider any additional feedback.

2. Autoxidation catalyzed by Co/Mn/Zr/Br in acetic acid has been reported for the conversion of lignin model compounds and even with commercially lignin from Sigma-Aldrich to aromatic monomers (DOI: 10.1002/adsc.200800614). The authors should explain the different reaction mechanisms and explain the high efficiency of their Mn/Zr catalyst.

We appreciate the opportunity to clarify these points and concerns of novelty.

First, we emphasize that in the originally submitted version of the paper, we cited and explicitly mentioned both of the Partenheimer studies in the Introduction of our paper (references 31 and 32). We have also added a reference of Co/Br/H₂O₂ oxidation of lignin model compounds by Clatworthy *et al.*

The primary distinction between the referenced work and our research is that we are focused on C–C cleavage reactions. In the Partenheimer studies, **Fig. A**, the aims of the model compound work were (1) to determine kinetic parameters for the oxidation of 3,4-dimethoxytoluene to veratric acid, and (2) to study

rates of the *in situ* phenol acetylation versus phenol oxidation leading to monomer loss. Aim (2) in his previous work was targeted because the lignin substrates contained β -O-4 bonds that produce phenolic products.

Fig. A. Illustration of the aims (1) and (2) of the Partenheimer Co/Mn/Br-mediated autoxidation of lignin and models.

Regarding the Clatworthy *et al.* reference, like Aim (2) above, the Co/Br/H₂O₂-mediated oxidation studied therein was also performed on β -O-4 lignin models. These studies target monomer production from C–O bond cleavage, and like Partenheimer’s work described above, the phenols generated as products inhibit the oxidation catalysis. Another difference is that O₂ is not used in these studies.

Taken together, the aforementioned reports did not systematically study C–C bond cleavage of lignin. Both report on the production of phenolic products that inhibit oxidation from the β -O-4 containing substrates. In our case, we are studying catalytic C–C bond cleavage using a substrate free of β -O-4 bonds and phenols and hence, with phenol stabilization, we do not encounter phenol inhibition.

We would like to again point out there is only one similar study, by Samec *et al.*, which presents the superstoichiometric oxidation of phenolic RCF dimers and oligomers using a TEMPO⁺ oxidant to produce 2,6-dimethoxybenzoquinone. In contrast, our system achieves similar yields but uses catalytic aerobic oxidation conditions. This approach thus represents a notable advance.

Regarding mechanistic differences between Co/Mn/Br/Zr and Mn/Zr, one obvious difference is the presence of the Br cocatalyst in Co/Mn/Br/Zr. This could impact initiation of the radical-chain mechanism, as shown in **Fig. B** directly below. With Br, initiation occurs through H-atom transfer chemistry (HAT). For Mn/Zr, we postulate that initiation could occur through single-electron transfer (SET) from the electron rich aromatics to Mn(III).

Fig. B. Differences in the initiation between (top) Co/Mn/Br catalyst with Br as the initiation cocatalyst (bottom) and initiation by a Mn/Zr catalyst where Mn(III) initiates the radical chain mechanism

To help address any concerns of mechanistic details, we have provided a more detailed description of the various mechanistic studies in the discussion section of our manuscript. This includes details regarding Zr, which can also be found in response to comment 5 from Reviewer 1, and those of Br, also included below. We hope our comments and modifications to the manuscript address the concerns of the reviewer.

The Br cocatalyst in Co/Mn/Br also differs significantly from our Mn/Zr catalyst. In the former system, Br initiates the radical chain process by H-atom transfer. In the absence of Br, it has been proposed that Mn(III) generates radicals by single-electron transfer from the electron-rich lignin aromatics via loss of a proton from a radical cation intermediate.⁷⁷ It has been suggested that Co(III) is generated from Co(II) in Co/O₂/AcOH systems by adventitious peroxide.⁷⁸

3. The units for catalyst loading and reaction yield are inconsistent.

The use of mol% for reporting catalyst loadings and reaction yields with model substrates ensures consistency in catalyst loadings, despite differences in molecular weights. However, when dealing with lignin, where accurately measuring the total aromatic content is challenging, we employ wt% but maintain the Mn/Zr molar ratio reporting used in the model studies. We firmly believe this approach is the most suitable to report catalyst loadings and yields in the context of our research. We note that this approach reflects that used by Samec *et al.* in their important work on C–C cleavage in lignin. We would like to retain consistency with their approach for ease of comparison.

4. The manuscript discusses the stability of products in standard reactions, can the author explain the relation between the stability of products and the main point of their work? Is there a way to increase the stability and total reaction yield of the aromatic monomers?

The relationship between the stability of products and the main point of our work – monomer production through C–C bond cleavage – is that degradation of our products occurs concurrent with monomer production. There is a time window from about 0.5 h to 3 h where there is no apparent net loss of monomers due to seemingly equal contributions from these competing phenomena. We found in our preliminary studies that longer reaction times (overnight), however, lead to significant losses in the overall yield, presumably due to a greater degree of monomer degradation over production. Regarding the second point on increasing the stability and total reaction yield, flow-based reactions are slated for our work to isolate monomers from the reaction as they are formed, thereby reducing the effect of competitive product degradation, but this effort is outside the scope of the current study.

5. The authors suggest more electron donating groups on phenol may increase the reaction yield, this should be demonstrated. Could the introduction of other kinds of protecting groups increase the reaction yield?

Thank you for the opportunity to clarify our statement. On page 3, we state: “*We hypothesized changing the phenol-stabilization group to one that is more electron donating would enable the Mn catalyst to oxidize the lignin model substrate*”.

We emphasize this does not imply greater product yield. Instead, it pertains to the feasibility of the Mn catalyst to facilitate oxidation of the lignin model substrate through the introduction of the methyl protecting group, which results in a more donating methoxy substituent on the arene relative to acetoxy group for the acetyl-protected phenols. We believe that introducing additional protecting groups beyond the comparison of acetyl and methyl protection could potentially divert focus from the primary message of this work, which aims to provide a proof-of-concept for the capacity of catalyzed autoxidation to cleave C–C bonds in lignin and produce bio-available monomer products for subsequent upgrading.

6. This work only used methylated lignin oil as a substrate, why not use native or commercially available lignin polymer, which contains less phenolic hydroxyl groups and would be better to utilize?

We reemphasize that our focus here is to study the cleavage of C–C bonds in lignin-derived materials. RCF oil oligomers are linked solely through C–C bonds, making RCF oil an ideal substrate to achieve this objective. Both native and commercial lignins contain C–O bonds, and phenolic monomers would be produced during autoxidation, as demonstrated in Partenheimer’s work (see response to comment 2). These phenols would act as antioxidants and hinder further oxidation.

In our work, we deliberately chose to use a fully methylated RCF oil free of phenol moieties and devoid of C–O bonds that could yield phenols. By using this substrate, we were able to overcome the potential

generation of phenols, thereby achieving higher yields than would have been possible with native or commercial lignins.

7. For the oil methylation process, some S-type monomers are difficult to be methylated, while no residual phenols in the methylated oils were detected according to the ³¹P NMR result, why?

We did not see any remaining phenolic S-type monomers. This could be due to our methylation procedure and work up, which is done by stirring the lignin oil with a large excess of methyl iodide and potassium carbonate for several days at room temperature. Further, and to ensure there are no residual phenols in our product, we do not acidify the mixture to quench the excess potassium carbonate. Thus, the aqueous layer during our liquid-liquid extraction is basic enough to keep residual phenoxides in the aqueous fraction of the extraction. With poplar containing S-type aromatics, 2 monomers were not fully recovered as their methylated derivatives. It is possible that these 2 S-type monomers do not get methylated fully, leaving behind their phenoxide forms that end up in the aqueous fraction during our extraction. We hope we have addressed this ambiguity.

Our methylation procedure, described in the SI, is copied here for ease of reference:

“Methylation of poplar RCF oil. A 250 mL round-bottom flask was charged with poplar RCF oil (7.55 g), potassium carbonate (30.0 g, 0.211 mol), acetonitrile (125 mL), and a stir-bar. Methyl iodide (61.0 g, 0.441 mol) was added, and the solution was stirred at room temperature. After 60 h, water (200 mL) was added to the resultant brown slurry, and the mixture was extracted with ethyl acetate (3 x 150 mL). The organic layers were combined and washed with water (2 x 100 mL) and brine (1 x 100 mL). The organic layer was subsequently dried with sodium sulfate, filtered, and the volatiles were removed by rotary evaporation leaving a brown viscous oil (9.13 g).”

8. Bacterial conversion of lignin derived aromatics to muconic acid has already been published by the same authors (<https://doi.org/10.1016/j.ymben.2021.12.010>), and the main results of bacterial conversion is not reported paper. It is not clear what improvement is reported in this manuscript.

The work referenced by the reviewer is indeed an example of bioconversion of lignin-related aromatics to muconic acid in *Pseudomonas putida* (of multiple such studies), but it differs substantially in its scope and findings compared to the current work. Namely, in Kuatsjah *et al.*, the focus was to improve a key enzymatic bottleneck in the utilization of *p*-coumaric acid (namely, hydroxylation of 4-hydroxybenzoate to protocatechuate) by combining biochemistry, structural biology, and systems biology. Therein, we translated the mechanistic findings to generate the engineered *P. putida* strain CJ781, which produced muconic acid from *p*-coumaric acid.

Unlike the present study, where we utilized a heterogeneous stream of lignin monomers derived from pine, Kuatsjah *et al.* used only the pure model compound (*p*-coumaric acid) as a substrate for muconic acid production in strain CJ781. Additionally, the *P. putida* metabolic pathway for bioconversion in the present study incorporates several new components, relative to strain CJ781. The methylation of pine RCF generated a slate of aromatic compounds with ring substituents not typically observed in lignin and not natively catabolized by *P. putida* (namely, *p*-methoxy groups; see **Supplementary Fig. 36**), so we engineered new catabolic pathways into *P. putida* to enable the demethylation of 3,4-dimethoxybenzoic acid (veratric acid) and 3,4-dimethoxybenzaldehyde (veratraldehyde). Heterologous expression of the CYP199A4 system, as well as *ivaAB*, in *P. putida* strain ACB263 therefore represents a substantial modification of strain CJ781 to accommodate more diverse aromatic substrates.

The reviewer also notes that the “main results of the bacterial conversion is not reported” – we are unsure what this means, but we are happy to consider further changes if the reviewer has specific suggestions. In addition to **Fig. 9**, the reviewer may find it helpful to review **Supplementary Figs. 36-40** and **Supplementary Tables 21-22** for more details on bacterial conversion of lignin monomers derived from pine.

9. Since the whole process is complicated, the mass balance of the total process needs to be provided.

Full mass balances for pine and poplar are calculated as follows: Pine: 20 wt% C–C cleavage monomers pine x (0.4 g pine oligomers from distillation/methylated pine RCF) x (8.381 g Methylated poplar RCF/7.869 Phenolic poplar RCF) x (0.465 g extracted pine lignin/ total lignin in pine biomass) = 4 wt% C–C cleavage monomers pine; Poplar: 19 wt% C–C cleavage monomers poplar x (0.43 g pine oligomers from distillation/methylated poplar RCF) x (9.131 g Methylated poplar RCF/7.547 Phenolic poplar RCF) x (0.79 g extracted poplar lignin/ total lignin in poplar biomass) = 8 wt% C–C cleavage monomers poplar.

We emphasize that the point of this work is to demonstrate proof-of-concept of a particular step and not to deliver a fully optimized process. To us, presenting mass balances does not seem useful unless all steps have been optimized from start to finish, and this was not our aim. Thus, we prefer to retain the mass balance of only the autoxidation step, as we have already done in eqs 1 and 2 of the manuscript.

Regarding an overall process, the overall yield could be improved by fully extracting lignin during RCF, retaining dimers during the distillation, and avoiding overoxidation reactions that lead to monomer degradation. We note we are actively pursuing high-yielding steps and process considerations informed by techno-economic analysis and life cycle assessment to be published in future work.

At this stage I am not convinced that the work offers sufficient novelty for publication in NC.

Reviewer #3:

The authors employ the Mn/Zr catalytic system to effectively cleave C-C bonds in lignin-derived dimers and oligomers obtained from pine and poplar via the autoxidation method. This process leads to the production of aromatic carboxylic acid, which can be further transformed into cis, cis-muconic acid using an engineered strain of *Pseudomonas putida* KT2440. The manuscript ingeniously combines chemical catalysis and biocatalysis in the conversion of renewable energy into fine chemicals, showcasing impressive scientific achievement. I recommend publishing this work in Nature Communications with minor revisions as outlined below:

We thank the reviewer for the positive and constructive feedback! We have attempted to address each comment below and in the revised manuscript.

1. The metabolic pathway of the engineered strain *P. putida* KT2440, responsible for the conversion of aromatic compounds into cis,cis-muconic acid, lacks sufficient clarity. I urge the authors to provide a detailed explanation in this regard.

We appreciate the reviewer's concern about this figure, and we have modified **Figure 9** and the text to further clarify the metabolic pathway. A previously published strain, *P. putida* CJ781, can convert vanillate to muconate (Kuatsjah *et al.*, *Metabolic Engineering*, 2022), and here, additional strain engineering was required to utilize the veratrate and veratraldehyde found in oxidation streams of methylated aromatic compounds:

(1) the CYP199A4 system was required to *p*-demethylate veratrate, and (2) the *ivaAB* monooxygenase was required to *p*-demethylate isovanillate.

We referenced strain CJ781 in the text and in **Supplementary Table 20**, and we encourage readers to review these for detailed information about the genotype and pathway. Additionally, we have now included a reference to strain CJ781 in the caption of **Figure 9**, and we have modified the figure to indicate which enzymatic steps were newly engineered in strain ACB263.

2. On page 6, paragraph 3, line 1, the statement "Optimization of reaction conditions led to a 28 mol% increase in RCF monomer yields" requires a more comprehensive explanation of the methodology employed. The current description is overly simplistic.

We thank the reviewer for pointing out the lack of clarity of our subheading. We have now updated the subheading to read "Autoxidative C–C cleavage affords monomers at 28 mol% of the RCF yields" and hope this change more clearly highlights the takeaway of this section.

3. The authors should discuss the distinguishing catalytic mechanisms between the Mn/Zr catalytic system and the Co/Mn/Br system. Although the author speculates on several models in paragraph 3 of page 11, they fail to validate their applicability to the Mn/Zr catalytic system. If relevant, I suggest including a detailed discussion of these catalytic mechanisms.

Validating the various models is a challenging task that is beyond the scope of this work, but we hope to pursue in future work. Nonetheless, in light of this reviewer comments, we have restructured the Discussion section to include a more detailed presentation of mechanistic hypotheses and discussion. This section can also be found in the response to comment 5 by Reviewer 1. We hope the reviewer finds this discussion useful.

4. On page 3, paragraph 4, line 4, please revise the phrase "(S)-type 4-propylsyringol" to "(S)-type 4-propylsyringol."

We would be happy to make a minor modification like this, but we would like to keep the comma after "syringol" rather than end the sentence with a period, in order to keep "(S)-type 4-propylsyringol, as shown in Fig 3.", as written for clarity.

5. In reference 47, please correct "cis, cis" to "*cis, cis*."

This has been done.

6. The concept of biological funneling is relatively novel. I recommend briefly summarizing it in the appropriate section of the introduction.

We thank the reviewer for this suggestion. We have added a description of biological funneling along with references in the last paragraph of the Introduction on page 2 to familiarize the reader with this concept.

7. Within the discussion section, I suggest emphasizing innovative summaries while reducing the focus on addressing existing problems. This adjustment will better highlight the novelty of this research.

We note that this was a topic of substantial discussion among the co-authors, and the two senior authors on this work acknowledge that they have different stylistic approaches to this. Considering this comment, we have now added a paragraph in the Discussion section highlighting the innovative aspects of this work, including C–C scission by thermolyzing radical intermediates and the engineering of *P. putida* to catabolize our methylated non-phenolic products. The paragraph is also included below.

While oxidations of lignin models and lignin have been studied with Co/Mn/Br and Co/Br,^{31, 32, 50} the target of those studies was not C–C bond cleavage. Rather, the goal was to increase monomer yields from oxidation of β -O-4-containing substrates through the in-situ acetylation of the phenolic products prior to their degradation under the oxidation conditions. Conversely, our study presents a method to overcome the inherent limit of monomer yields from C–O cleavage alone by catalytically cleaving C–C bonds in lignin oligomers. This is accomplished by thermolyzing high energy radical intermediates, ultimately effecting C–C cleavage through β -scission reactions. We further describe methods that enable the identification and quantification of a mixture of oxygenated products along with their biological conversion using engineered strains of *Pseudomonas putida*.

Regarding the section on limitations, we have now condensed this information into a single paragraph. We hope the reviewer finds value in our modifications.

Reviewer #4:

In this study, the authors extended their tandem approach, previously demonstrated in their Science paper, to leverage lignin valorization. The limited utilization efficiency of lignin oligomers remains a significant obstacle within the lignin community, leading research efforts to yield results that may not be immediately ideal but represent essential advancements towards practical lignin utilization. Consequently, this study addresses a critical challenge in the depolymerization of lignin oligomers to lignin monomers through catalytic cleavage of C-C bonds. Additionally, the authors candidly acknowledge the current limitations of their proposed approach. As a researcher and reviewer within the related fields, I appreciate this honesty, which, in my understanding, does not diminish the novelty of the study but provides readers with valuable insights to construct a comprehensive understanding of this investigation.

We appreciate the positive feedback from the reviewer! We have attempted to address each of these comments below.

Main concerns:

1. The authors reference an unpublished work of their own, which is generally acceptable; however, it is essential to highlight the distinctions between the two studies. At first glance, based solely on the titles, the two studies may appear quite similar. If the primary differentiation lies only in the oxidation catalysts employed for lignin depolymerization, the novelty of the findings may be deemed insufficient for publication in Nature Communications.

We appreciate this comment from the reviewer. Although we included the rationale of this work in paragraphs 4 and 5 of the Introduction, this comment indicates that we did not make this distinction sufficiently clear. We have now included an additional sentence in paragraph 1 of the Results section, which we feel helps differentiate our work from our Co/Mn/Br/OAc companion study. We have also explained the differences below for clarity.

Fig. C. Illustration of the differences between this work, Mn/Zr on methyl-protected lignin oils, and the Co/Mn/Br companion study on acetyl-protected poplar.

It is widely acknowledged that it is exceptionally difficult to make lignin conversion to monomers economically viable. As a step toward this goal, our work addresses the potential cost saving alternative of a Mn/Zr catalyst, as opposed to the Co/Mn/Br catalyst in our companion work.

While manganese is known to be a less active autoxidation catalyst, we establish its viability for lignin C–C cleavage when methylation is used for phenol protection as opposed to the acetyl protection of the Co/Mn/Br study. Unexpectedly, the manganese catalyst also affords higher yields of oxidation products. We attribute this to the difficulty in demethylating the monomers, which would lead to reactive phenol products that could undergo undesirable conversion to other compounds.

However, methylation poses a new challenge, as it is troublesome to upgrade the oxygenated products due to the difficulty in cleaving aryl ether bonds compared to the aryl ester bonds in the acetyl protection case. To overcome this issue, we present a metabolic engineering approach that enables us to demethylate the

aromatics and further convert them through decarboxylation and oxidative ring opening to *cis,cis*-muconic acid.

These unique aspects of our work distinguish it from our companion Co/Mn/Br/acetyl-protection study in review. We believe the insights gained are unique and complementary and, thus, well-suited for publication in *Nature Communications*.

2. In the tandem approach towards practical conversion of lignin oligomers, multiple steps are unfortunately omitted or oversimplified.

Indeed, there are multiple steps taken to prepare our model substrate used in this work. However, we acknowledged this in paragraph 3 (now paragraph 2) of the Discussion section, and we also note in paragraph 3 of the Introduction that we use RCF only as a convenient means to obtain a lignin substrate free of β -O-4 bonds for a systematic study of C–C bond cleavage.

To provide further clarification, we have now included a graphical overview of the steps in Supplementary Fig 1.

2.1 The use of MeI as the stabilization reagent may give a poor impression that the synthetic route was designed for an organic chemistry lab. While advancements in synthetic chemistry for lignin depolymerization are appreciated, the lignin community, presently interested in practical lignin utilization, tends to favor valorization pathways illustrated with sufficient proof of viability in real lignin valorization. In this case, dimethyl carbonate is proposed as an alternative, but the references (48 and 49) provided are insufficient to validate the suitability of the reaction for lignin methylation.

We thank the reviewer for pointing this out. We have added the reference 10.1039/C4GC01759E, which highlights prospects for the use of dimethyl carbonate as a more compelling alternative methylating agent.

2.2 The oxidative depolymerization of lignin oligomers was conducted in acetic acid at a relatively low substrate loading (4.3 mg/mL). According to the findings in Fig. 7d, the product yield increased with higher substrate loading. To gain more practical insights, it is highly recommended to explore the depolymerization reaction at higher substrate loadings, as converting 4.3 g of oligomer samples in 1 L acetic acid would barely be considered in real cases.

We recognize the significance of exploring greater substrate loadings to simulate conditions more representative for industrial application. Our choice to work with a loading of 4.3 mg/mL stemmed from the challenges we encountered with oxygen loading in our 75 mL Parr vessel relative to the number of cleavable C–C bonds and when trying to scale up the preparation of our oligomer substrate using our current laboratory equipment. When the partial pressure of oxygen is not high enough, coupling of carbon–carbon radicals occurs and lowers the overall yield. Moreover, it is difficult to prepare enough substrate to optimize yields with high substrate loadings. This is because our distillation apparatus for this study was operating at its upper limit, and attempts to employ larger glassware for the distillation did not achieve the same degree of monomer separation and recovery. The amount of oligomer substrate we prepared is significantly greater than the state-of-the-art. To provide relevant context, the amount of oligomer we synthesized is 12x (pine) and 14x (poplar) that produced by Samec and coworkers in their 2021 report in *Nature Chemistry* (10.1038/s41557-021-00783-2).

To maintain consistency in our results, we also believed it was important to perform all reactions using the same oligomer batch. This assured the products depicted in **Figs. 7** and **8** were directly generated from the substrate characterized in **Supplementary Figs. 30-35**. Given these parameters, we operated at the lowest viable substrate loadings, using just enough acetic acid to keep our 75 mL reactor thermocouple submerged to allow proper temperature regulation. Our loadings also ensure super stoichiometric oxygen relative to the number of cleavable C–C bonds, thereby avoiding possible C–C radical coupling reactions.

Although **Fig. 7d** suggests that increased product yield might be achieved at higher substrate loadings, we contend that our current loadings, given the aforementioned constraints, still provide valuable insights into the reaction mechanisms and product profiles. Furthermore, we believe our findings do not compromise our central thesis: that autoxidation can effectively cleave the C–C bonds of lignin, yielding bioavailable monomers convertible to a singular product.

2.3 The process of biological funneling using the depolymerized oligomers is not clearly explained. One crucial step is omitted, in which additional details are required to elucidate how the depolymerized oligomers were isolated from the mixture after the oxidation reaction. According to the supplementary experimental information (PS26), the samples (~30 mg in 45 mL acetate acid) were dried under nitrogen at -20 °C. The rationale behind this operation should be carefully elucidated, as it serves as a pivotal step bridging the chemical oxidation and biological funneling processes.

We thank the reviewer for pointing out the lack of clarity here. We rewrote the “Preparation of bacterial culture media” section and hope it is clearer.

References

1. Steinmetz GR, Sumner CE. Role of amine promoters in cobalt/bromide-catalyzed oxidations. *Journal of Catalysis* **100**, 549-551 (1986).
2. Steinmetz G, Lafferty N, Sumner Jr C. The cobalt/zirconium-catalyzed oxidation of cyclohexane to adipic acid. *Journal of Molecular catalysis* **49**, L39-L42 (1988).
3. Baxendale J, Wells C. The reactions of Co (III) with water and with hydrogen peroxide. *Transactions of the Faraday Society* **53**, 800-812 (1957).
4. Hendriks CF, van Beek HC, Heertjes PM. Reactions of some peracids and hydroperoxides with cobalt (II) and cobalt (III) acetate in acetic acid solution. *Industrial & Engineering Chemistry Product Research and Development* **18**, 38-43 (1979).
5. Sumner Jr CE. Interconversion of dinuclear and oxo-centered trinuclear cobaltic acetates. *Inorganic Chemistry* **27**, 1320-1327 (1988).
6. Sumner Jr CE, Steinmetz GR. Isolation of oxo-centered cobalt (III) clusters and their role in the cobalt bromide catalyzed autoxidation of aromatic hydrocarbons. *Journal of the American Chemical Society* **107**, 6124-6126 (1985).
7. Chavan S, Halligudi S, Srinivas D, Ratnasamy P. Formation and role of cobalt and manganese cluster complexes in the oxidation of p-xylene. *Journal of Molecular Catalysis A: Chemical* **161**, 49-64 (2000).
8. Babushkin D, Talsi E. Multinuclear NMR spectroscopic characterization of Co (III) species: Key intermediates of cobalt catalyzed autoxidation. *Journal of Molecular Catalysis A: Chemical* **130**, 131-137 (1998).
9. Jones GH. p-Xylene autoxidation studies. Oxidation of cobalt (II) and manganese (II) acetates by peracids. *Journal of the Chemical Society, Chemical Communications*, 536-537 (1979).
10. Partenheimer W. Methodology and scope of metal/bromide autoxidation of hydrocarbons. *Catalysis Today* **23**, 69-158 (1995).
11. Lande SS, Falk CD, Kochi JK. Cobalt (III) acetate from the ozonation of cobaltous acetate. *Journal of Inorganic and Nuclear Chemistry* **33**, 4101-4109 (1971).
12. Jiao X-D, Espenson JH. Kinetics of manganese (III) acetate in acetic acid: Generation of Mn (III) with Co (III), Ce (IV), and dibromide radicals; Reactions of Mn (III) with Mn (II), Co (II), hydrogen bromide, and alkali bromides. *Inorganic Chemistry* **39**, 1549-1554 (2000).
13. Chester AW, Scott EJ, Landis PS. Zirconium cocatalysis of the cobalt-catalyzed autoxidation of alkylaromatic hydrocarbons. *Journal of Catalysis* **46**, 308-319 (1977).
14. Partenheimer W. The effect of zirconium in metal/bromide catalysts during the autoxidation of p-xylene: Part i. Activation and changes in benzaldehyde intermediate formation. *Journal of Molecular Catalysis A: Chemical* **206**, 105-119 (2003).
15. Tomas RA, Bordado JC, Gomes JF. p-Xylene oxidation to terephthalic acid: a literature review oriented toward process optimization and development. *Chemical Reviews* **113**, 7421-7469 (2013).
16. Scott E, Chester A. Kinetics of the cobalt-catalyzed autoxidation of toluene in acetic acid. Role of cobalt. *The Journal of Physical Chemistry* **76**, 1520-1524 (1972).
17. Blake AB, Chipperfield JR, Lau S, Webster DE. Studies on the nature of cobalt (III) acetate. *Journal of the Chemical Society, Dalton Transactions*, 3719-3724 (1990).

18. Chipperfield JR, Lau S, Webster DE. Speciation in cobalt (III)-catalysed autoxidation: Studies using thin-layer chromatography. *Journal of Molecular Catalysis* **75**, 123-128 (1992).
19. Partenheimer W. The effect of zirconium in metal/bromide catalysts on the autoxidation of p-xylene. *Journal of Molecular Catalysis A: Chemical* **206**, 131-144 (2003)
20. Partenheimer W, Grushin VV. Synthesis of 2, 5-Diformylfuran and Furan-2, 5-Dicarboxylic Acid by Catalytic Air-Oxidation of 5-Hydroxymethylfurfural. Unexpectedly Selective Aerobic Oxidation of Benzyl Alcohol to Benzaldehyde with Metal= Bromide Catalysts. *Advanced Synthesis & Catalysis* **343**, 102-111 (2001).
21. DiCosimo R, Szabo HC. Oxidation of lignin model compounds using single-electron-transfer catalysts. *The Journal of Organic Chemistry* **53**, 1673-1679 (1988).
22. Partenheimer W. Characterization of the reaction of cobalt (II) acetate, dioxygen and acetic acid, and its significance in autoxidation reactions. *Journal of Molecular Catalysis* **67**, 35-46 (1991).

REVIEWER COMMENTS

Reviewer #2 (Remarks to the Author):

The authors have improved the manuscript, but I have some further comments as my original comments were apparently not clear to the authors or not fully addressed.

1. Paragraphs 2-3 of the introduction on C–C bond cleavage is confusing. C–C bond cleavage under very different conditions (hydrogenolysis/aerobic oxidation/stoichiometric oxidants, lignocellulose/native lignin/lignin oil) are not comparable. While mentioning different C–C bond cleavage methods may be helpful, when carefully described, a detailed focus on C–C bond cleavage of lignin oil would be helpful.
2. From previous reports, the advantage of this work is the transformation of methylated lignin oil (not lignin oil) without Co. Many homogeneous catalysts (ACS Catal. 2013, 3, 3111-3122), heterogeneous catalysts (Angew. Chem. Int. Ed. 2020, 59, 19268-19274; J. Am. Chem. Soc. 2021, 143, 15462-15470), or even metal free catalysts (Angew. Chem. Int. Ed. 2023, 62, e202219217; Green Chem. 2018, 20, 170-182; Nature Chem. 2019, 11, 71-77) have the SET and HAT capacity for C–C bond cleavage of natural and artificial polymers. Hence, the mechanism is the most interesting part and further experiments comparing Mn/Zr with Co/Mn/Zr/Br would be useful, for example radical trapping, radical clock experiment, etc.
3. I suggest moving the stability part of products to supporting information.

Reviewer #3 (Remarks to the Author):

The author has solved my question, and I have no further questions.

Reviewer #4 (Remarks to the Author):

I would like to recommend the revised manuscript for publishing since the vast majority of the previous referees' concerns were properly addressed.

Editorial note: Reviewer #4 has assessed author's responses to Reviewer #1 and considers them addressed, and supports publication.

Reviewer #2 (Remarks to the Author):

The authors have improved the manuscript, but I have some further comments as my original comments were apparently not clear to the authors or not fully addressed.

1. Paragraphs 2-3 of the introduction on C–C bond cleavage is confusing. C–C bond cleavage under very different conditions (hydrogenolysis/aerobic oxidation/stoichiometric oxidants, lignocellulose/native lignin/lignin oil) are not comparable. While mentioning different C–C bond cleavage methods may be helpful, when carefully described, a detailed focus on C–C bond cleavage of lignin oil would be helpful.

We appreciate the referee's continued attention to the presentation of examples of lignin C–C cleavage in the Introduction. To clarify, we are not trying to "compare" different conditions for C–C cleavage, but rather to highlight a few key systems where C–C cleavage seems to be operative. We note that it can be difficult to discern between C–C and C–O cleavage in many lignin depolymerization systems, because the monomer yields are often below the theoretical yield that can be obtained through C–O cleavage alone. Since there is a general dearth of well-defined C–C bond cleavage studies in the literature, we have elected to highlight a few specific relevant examples, without discriminating based on the conditions (e.g., reductive, oxidative, or other). We appreciate the reviewer's inclusion of some specific references in comment #2 below. We now include some of these in our commentary in the updated second paragraph.

Paragraph 3 of the Introduction is intended to provide a more detailed summary of the one system that does exist, focused on C–C cleavage in lignin oil. To our knowledge, this work by Samec *et al.* is the only study reported to date that directly explores C–C cleavage in RCF lignin oil, and it represents the state of the art. We include a separate paragraph to highlight the results of this system because it is the primary basis for comparison with our work.

We hope the reviewer finds the modified text provides adequate introductory context for the present study.

2. From previous reports, the advantage of this work is the transformation of methylated lignin oil (not lignin oil) without Co. Many homogeneous catalysts (ACS Catal. 2013, 3, 3111-3122), heterogeneous catalysts (Angew. Chem. Int. Ed. 2020, 59, 19268-19274; J. Am. Chem. Soc. 2021, 143, 15462-15470), or even metal free catalysts (Angew. Chem. Int. Ed. 2023, 62, e202219217; Green Chem. 2018, 20, 170-182; Nature Chem. 2019, 11, 71-77) have the SET and HAT capacity for C–C bond cleavage of natural and artificial polymers. Hence, the mechanism is the most interesting part and further experiments comparing Mn/Zr with Co/Mn/Zr/Br would be useful, for example radical trapping, radical clock experiment, etc.

We thank the reviewer for the references here, some of which are relevant to C–C cleavage in lignin. As noted above, several of these are now included in our revised manuscript, prioritizing those directly focused on C–C cleavage in lignin.

We also value the reviewer's interest in the mechanistic aspects of this work. Mechanistic studies on Co/Mn/Zr autoxidation systems have been performed for decades due to the industrial interest in this chemistry for conversion of *p*-xylene to terephthalic acid, and the results of these studies have been reviewed extensively by Partenheimer (see ref 28 in the revised manuscript and references 1-11 within this article) and more recently by Adamian and Gong (ref 31 in the revised manuscript). We cite these reviews and additional references from this mechanistic literature in the present submission (see refs 16, 17, 21, 32, and 59). Refs 67-84, added in the previous revision are noteworthy because they provide mechanistic insights specifically relevant to Zr in the Mn/Zr catalyst system used in our study.

As the general types of radicals formed during Mn- and Co/Mn-catalyzed autoxidation are well-known, and the routes through which they are generated have been extensively outlined in the aforementioned review and primary articles, we elected not to do additional experiments here (radical trapping, radical clock). Such experiments could certainly be performed (at considerable time and expense), but would inevitably feature model systems that only indirectly relate to the lignin oil under investigation here (e.g., the radical clock experiments), and/or would generate products susceptible to further oxidation under the reaction conditions that would be highly complicated to analyze and characterize from the complex reaction mixture. Because

the extensive prior literature has clearly documented the (catalytic) radical-chain nature of these reactions, we do not expect additional mechanistic experiments to offer new insights.

3. I suggest moving the stability part of products to supporting information.

We thank the reviewer for their suggestion. The product stability studies address an important complication that arises from the batch reactor setups used here. Inclusion of this content in the body of the manuscript not only provides transparency on the limitations of our existing reaction system, but it also suggests that better performance should be possible under continuous process conditions that will be the focus of future studies. For these reasons, we would prefer to retain this content in the body of the manuscript.

REVIEWERS' COMMENTS

Reviewer #2 (Remarks to the Author):

I am satisfied with the response/changes made by the authors and recommend publication.